# Lung tropism in hospitalized patients following infection with SARS-CoV-2 variants from D614G to Omicron BA.2

Yosuke Hirotsu [1,6✉], Yumiko Kakizaki[2,6], Akitoshi Saito[3], Toshiharu Tsutsui[2], Syunya Hanawa[2], Haruna Yamaki[2], Syuichiro Ide[2], Makoto Kawaguchi[2], Hiroaki Kobayashi[2], Yoshihiro Miyashita[2] & Masao Omata[4,5]

## Abstract

**Background** The genetic and pathogenic characteristics of SARS-CoV-2 have evolved from the original isolated strains; however, the changes in viral virulence have not been fully defined. In this study, we analyzed the association between the severity of the pathogenesis of pneumonia in humans and SARS-CoV-2 variants that have been prevalent to date.

**Methods** We examined changes in the variants and tropism of SARS-CoV-2. A total of 514 patients admitted between February 2020 and August 2022 were included and evaluated for pneumonia by computed tomography (CT) as a surrogate of viral tropism.

**Results** The prevalence of pneumonia for each variant was as follows: D614G (57%, 65/114), Alpha (67%, 41/61), Delta (49%, 41/84), Omicron BA.1.1 (26%, 43/163), and Omicron BA.2 (11%, 10/92). The pneumonia prevalence in unvaccinated patients progressively declined from 70% to 11% as the variants changed: D614G (56%, 61/108), Alpha (70%, 26/37), Delta (60%, 38/63), BA.1.1 (52%, 15/29), and BA.2 (11%, 2/19). The presence of pneumonia in vaccinated patients was as follows: Delta (16%, 3/19), BA.1.1 (21%, 27/129), and BA.2 (11%, 8/73). Compared with D614G, the areas of lung involvement were also significantly reduced in BA.1.1 and BA.2 variants.

**Conclusions** Compared with previous variants, there was a marked decrease in pneumonia prevalence and lung involvement in patients infected with Omicron owing to decreased tropism in the lungs that hindered viral proliferation in the alveolar epithelial tissue. Nevertheless, older, high-risk patients with comorbidities who are infected with an Omicron variant can still develop pneumonia and require early treatment.

## Plain language summary

The SARS-CoV-2 virus changes over time with the differing viruses described as variants. The different variants of SARS-CoV-2 have an impact on how easily they infect people and the effects they have on infected individuals. Here, we examined images of the lungs of patients hospitalized with COVID-19 to investigate whether they had pneumonia, a type of swelling in the lung. Compared with the variant found early in the pandemic, the more recent Omicron variant led to a decreased rate of pneumonia in infected individuals. Our findings emphasize the need for early treatment, as pneumonia may progress in older patients or those with other illnesses.

[1] Genome Analysis Center, Yamanashi Central Hospital, 1-1-1 Fujimi, Kofu, Yamanashi, Japan. [2] Lung Cancer and Respiratory Disease Center, Yamanashi Central Hospital, 1-1-1 Fujimi, Kofu, Yamanashi, Japan. [3] Department of Radiology, Yamanashi Central Hospital, 1-1-1 Fujimi, Kofu, Yamanashi, Japan. [4] Department of Gastroenterology, Yamanashi Central Hospital, 1-1-1 Fujimi, Kofu, Yamanashi, Japan. [5] The University of Tokyo, 7-3-1 Hongo, Bunkyo-ku, Tokyo, Japan. [6] These authors contributed equally: Yosuke Hirotsu, Yumiko Kakizaki. ✉email: hirotsu-bdyu@ych.pref.yamanashi.jp

Since its emergence in 2019, SARS-CoV-2 has caused an enormous number of infections and deaths. However, the population is becoming increasingly immune owing to an increase in the number of natural infections and increasing numbers of vaccinated individuals. In some areas, mass immunity has been achieved, and infection mitigation is gradually progressing[1–3]. In addition, therapeutics such as monoclonal antibodies, antiviral drugs, and anti-inflammatory drugs have been developed that have helped to reduce the risk of infection and the severity of COVID-19[4]. The virus has accumulated mutations over time[5], and the emergence of more infectious viral variants has led to a temporary increase in infections, but a decrease in viral virulence has also occurred[6]. Breakthrough infections can occur in individuals who have received more than two doses of vaccine[7–9]. Against this background, to improve infection control, we must clearly understand the infectivity and virulence of SARS-CoV-2 by analyzing the data that have accumulated over the past 2.7 years.

In Japan, national guidelines recommend admitting COVID-19 patients to the hospital even if their symptoms are mild enough to allow them to stay at home. Japan has a comprehensive national health insurance system that allows for chest computed tomography (CT) scans even in asymptomatic patients. Chest CT scanning can detect subtle pulmonary lesion changes, such as the peripherally distributed bilateral and multilobar ground-glass opacities, known as typical image of COVID-19 pneumonia. We previously performed whole-genome sequencing to determine genotype changes in SARS-CoV-2 over 2.7 years[10–13]. Data regarding the presence or absence of pneumonia in a wide range of clinical conditions have also accumulated.

In the current study, we investigated changes in lung viral tropism in COVID-19 patients by examining changes in the SARS-CoV-2 variants and the prevalence of pneumonia in vaccinated and unvaccinated patients. Results showed a higher risk of pneumonia with older age, male gender, and the Alpha variant, while being fully vaccinated and having the Omicron BA.2 variant were associated with a lower risk. The study also found differences in disease severity and outcomes between variants. These findings have important clinical implications for the management of COVID-19 and may inform vaccination and treatment decisions.

## Methods

**Patients**. This study included 730 hospitalized patients who tested positive for SARS-CoV-2 infection by PCR and/or antigen assay between February 2020 and August 2022. We excluded 84 hospitalized patients who did not undergo SARS-CoV-2 genomic analysis due to low viral loads. We also excluded 21 individuals who initially tested positive for SARS-CoV-2 by PCR but were likely false positives according to subsequent analyses[14]. A further 110 patients, including children or pregnant women who had not undergone CT scanning, were also excluded. Only one patient with the Gamma variant was detected during this period, but this case was removed from subsequent analyses because of the small number of Gamma variant detections[13].

In total, 514 hospitalized patients were finally included in this study. The CT images of the lungs were independently evaluated for the presence of pneumonia by a radiologist and two respiratory physicians. COVID-19-typical pneumonia was identified as peripherally distributed bilateral and multilobar ground-glass opacities, which are characteristic CT findings that have been reported in COVID-19 pneumonia[15,16]. Medical staff collected data on the medical condition, vaccination status, and disease course from patient interviews and retrospectively examined the electronic medical records. People who had received at least two vaccine doses with more than 14 days passing since the second dose were considered fully vaccinated. Breakthrough infection was defined as a positive SARS-CoV-2 PCR or antigen test result in a fully vaccinated individual.

The Institutional Review Board of the Clinical Research and Genome Research Committee at our hospital approved this study and the use of an opt-out consent method (approval no. C2019-30). The requirement for written informed consent was waived because this was an observational study, and there is an urgent need to collect COVID-19 data.

**SARS-CoV-2 diagnostic testing**. Multiple molecular diagnostic platforms of nucleic acid amplification and antigen testing were used to diagnose SARS-CoV-2 infection. The diagnostic tests used were reverse transcription PCR (in accordance with the protocol developed by the National Institute of Infectious Diseases in Japan[17]), the FilmArray Respiratory Panel 2.1 test with the FilmArray Torch system (bioMérieux, Marcy-l'Etoile, France)[18], the Xpert Xpress SARS-CoV-2 test using Cepheid GeneXpert (Cepheid, Sunnyvale, CA)[19], and the Lumipulse antigen test with the LUMIPULSE G600II system (Fujirebio, Inc., Tokyo, Japan)[20,21]. All tests were performed with material obtained from nasopharyngeal swabs immersed in viral transport medium (Copan, Murrieta, CA).

**SARS-CoV-2 whole-genome analysis**. Whole-genome sequencing analysis was conducted on SARS-CoV-2-positive samples collected from hospitalized patients as previously described[11]. In brief, SARS-CoV-2 genomic RNA was reverse-transcribed into cDNA and amplified using the Ion AmpliSeq SARS-CoV-2 Research Panel or the Ion AmpliSeq SARS-CoV-2 Insight Research Assay (Thermo Fisher Scientific, Waltham, MA) on the Ion Torrent Genexus system according to the manufacturer's instructions[11,13]. Sequencing reads were processed and quality was assessed using Genexus software with SARS-CoV-2 plugins. The sequencing reads were then mapped and aligned using the Torrent Mapping Alignment Program. After initial mapping, a variant call was performed using Torrent Variant Caller. The COVID19AnnotateSnpEff plugin was used to annotate the variants. Assembly was performed using Iterative Refinement Meta-Assembler[22].

The viral clade and lineage classifications were conducted using Nextstrain[23] and Phylogenetic Assignment of Named Global Outbreak Lineages[24]. Sequence data were deposited in the Global Initiative on Sharing Avian Influenza Data EpiCoV database[5].

**CT imaging**. All patients underwent CT immediately after arriving at the hospital. CT images (64 × 0.5 mm or 32 × 1.0 mm-detector row/automatic exposure control/120 kVp) were obtained with the Aquilion 64 or the Aquilion 64CX system (Canon Medical, Tochigi, Japan). The 0.5 mm or 1.0 mm-slice thickness high-resolution computed tomography images were reconstructed using the FC51 reconstruct function.

**CT image analysis**. All CT images were evaluated by an experienced radiologist specialized in thoracic radiology with 25 years of experience. Definitions of radiological terms are based on the standard glossary of terms for chest imaging reported by the Fleischner Society[25]. On the basis of previous reports, a pneumonia diagnosis was made and the main findings on chest CT were classified into three categories: consolidation (CON), crazy-paving appearance (CPA), and ground-glass opacity (GGO)[26–28]. The distribution of lung abnormalities was divided into three sections from the pulmonary hilum to the subpleural area and classified as those that were only subpleural (peripheral), those

that extended near to the pulmonary hilum (central), and those that extended to the middle area between the peripheral and central areas. Semi-quantitative CT scores were evaluated for the right upper lobe, right middle lobe, right lower lobe, left upper lobe, and left lower lobe based on previous reports[29]. Briefly, the CT scores were calculated according to the lesion area as follows: 0, 0% involvement; 1, <5% involvement; 2, 5%–25% involvement; 3, 26%–50% involvement; 4, 50%–75% involvement; 5, >75% involvement. The total CT score was the sum of the individual scores, ranging from 0 (no involvement) to 25 (maximum involvement).

**Serological analysis.** We measured the titers of anti-nucleocapsid (N) protein antibody and anti-spike (S) protein receptor-binding domain antibody using the Elecsys Anti-SARS-CoV-2 antibody test (Roche Diagnostics, Basel, Switzerland) on a cobas® 8000 automated platform[14]. This assay utilizes the electro-chemiluminescence immunoassay principle. For the anti-N antibody, samples with a cutoff index (COI) < 1.0 were considered negative, while those with a COI ≥ 1.0 were considered positive. For the anti-S antibody, samples containing <0.8 U/mL were considered negative, while those containing ≥0.8 U/mL were considered positive, in accordance with the manufacturer's instructions.

**Statistical analysis.** All statistical tests and visualizations were performed with R, version 4.1.1 (R Foundation for Statistical Computing) or RStudio (https://www.rstudio.com/). The following R packages were used for data cleaning, analysis, and visualization: ggplot2 (v3.3.5), dplyr (v1.0.7), tidyr (v1.1.3), patchwork (v1.1.1), gtsummary (v1.5.2), and flextable (v.0.7.0). A multivariate logistic regression analysis was performed to evaluate the effects of age (over or under 65 years), sex, viral lineage, and vaccination status on the likelihood of having COVID-19 pneumonia. Odds ratios (ORs) and the corresponding 95% confidence intervals (CIs) were calculated. $P$-values <0.05 were considered to be statistically significant.

**Reporting summary.** Further information on research design is available in the Nature Portfolio Reporting Summary linked to this article.

## Results

**Pneumonia prevalence and clinical background.** Patients admitted to our hospital between February 2020 and August 2022 were included in the study. Of the 730 hospitalized patients, 514 were evaluated for pneumonia by CT scanning, and patient samples were sequenced to identify the SARS-CoV-2 variant. Of these 514 patients (median age: 66.5 years [interquartile range (IQR), 46–81], female: 219, male: 295), 314 (61%) had no COVID-19-typical pneumonia, while 200 (39%) had COVID-19-typical pneumonia (Supplementary Data 1). Compared with patients without pneumonia, those with pneumonia tended to be male ($P = 8.6 \times 10^{-4}$), unvaccinated ($P = 8.0 \times 10^{-18}$), require supplemental oxygen ($P = 8.1 \times 10^{-76}$) and ventilator support ($P = 0.016$), and have worse outcomes ($P = 0.002$) (Supplementary Data 1). Compared with D614G, Alpha, and Delta variants, patients infected with Omicron BA.1.1 or BA.2 variants had significantly lower rates of pneumonia (D614G vs. BA.1.1, adjusted $P = 2.6 \times 10^{-6}$ D614G vs. BA.2, adjusted $P = 1.9 \times 10^{-11}$ Alpha vs. BA.1.1, adjusted $P = 2.3 \times 10^{-7}$ Alpha vs. BA.2, adjusted $P = 4.9 \times 10^{-12}$ Delta vs. BA.1.1, adjusted $P = 3.1 \times 10^{-3}$ Delta vs. BA.2, adjusted $P = 1.8 \times 10^{-7}$, pairwise Fisher's exact test) (Supplementary Table 1). Of the 38 breakthrough infections with pneumonia, remdesivir was administered in 29 (76%),

**Table 1 Multivariant logistic regression for factors associated with pneumonia.**

| Characteristic | OR | 95% CI | P-value |
|---|---|---|---|
| Age | | | |
| <65 (Reference) | — | — | |
| ≥ 65 | 3.38 | 2.02–5.83 | **6.4 × 10⁻⁶** |
| Sex | | | |
| Female (Reference) | — | — | |
| Male | 2.14 | 1.38–3.36 | **8.2 × 10⁻⁴** |
| Lineage | | | |
| D614G (Reference) | — | — | |
| Alpha | 2.39 | 1.05–5.71 | **0.042** |
| Delta | 1.31 | 0.70–2.47 | 0.4 |
| BA.1.1 | 0.80 | 0.39–1.66 | 0.6 |
| BA.2 | 0.20 | 0.08–0.46 | **3.0 × 10⁻⁴** |
| Vaccine status | | | |
| Unvaccinated (Reference) | — | — | |
| Two or more doses | 0.17 | 0.09–0.33 | **1.6 × 10⁻⁷** |

OR odds ratio, CI confidence interval.
Bold values indicate statistical significance $p<0.05$.

casirivimab/imdevimab in 1 (2.6%), sotrovimab in 2 (5.3%), and molnupiravir in 4 (11%) cases; nirmatrelvir/ritonavir were not used for treatment (Supplementary Table 2).

**Multivariable logistic regression analysis.** A multivariate logistic regression model was applied to examine independent predictors contributing to the presence or absence of COVID-19-typical pneumonia as determined by CT and their corresponding ORs and 95% CIs. The following factors were associated with a higher risk of COVID-19 pneumonia: 65 years and older (OR = 3.38; 95% CI, 2.02–5.83), being male (OR = 2.14; 95% CI, 1.38–3.36), and the Alpha variant (OR = 2.39; 95% CI, 1.05–5.71) (Table 1). The BA.2 variant (OR = 0.20; 95% CI, 0.08–0.46) and being fully vaccinated (OR = 0.17; 95% CI, 0.09–0.33) were associated with a lower risk of developing COVID-19 pneumonia (Table 1).

**Variants and CT pneumonia images.** The number of patients hospitalized with COVID-19 has intermittently increased over the past 2.7 years (Fig. 1a). The variants identified by whole-genome sequencing during the observation period were D614G ($n = 114$), Alpha ($n = 61$), Delta ($n = 84$), Omicron BA.1.1 ($n = 163$), and Omicron BA.2 ($n = 92$).

The CT findings of COVID-19-typical pneumonia were examined in detail. Pneumonia was observed in patients infected with the D614G (57%, 65/114), Alpha (67%, 41/61), Delta (49%, 41/84), Omicron BA.1.1 (26%, 43/163), and Omicron BA.2 (11%, 10/92) variants (Fig. 1b). There was a substantial decrease in the presence of pneumonia between the Alpha (67%) and Delta (49%) variants compared with the Omicron BA.1.1 (26%) and Omicron BA.2 (11%) variants (Fig. 1b).

**Variant, supplemental oxygen and ventilator use, and outcomes.** If a patient's arterial oxygen saturation decreased, the use of supplemental oxygen was initiated regardless of the cause. Supplemental oxygen was required for patients infected with the D614G (65%, 74/114), Alpha (72%, 44/61), Delta (49%, 41/84), Omicron BA.1.1 (39%, 63/163), and Omicron BA.2 (20%, 18/92) variants (Fig. 1c). The number of patients requiring supplemental oxygen decreased as the viral genotype successively changed. For those infected with Omicron BA.2, 20% of patients required supplemental oxygen, which was higher than the 11% of patients with pneumonia identified by chest CT. However, for eight of these patients, oxygen was supplied for other reasons (e.g.,

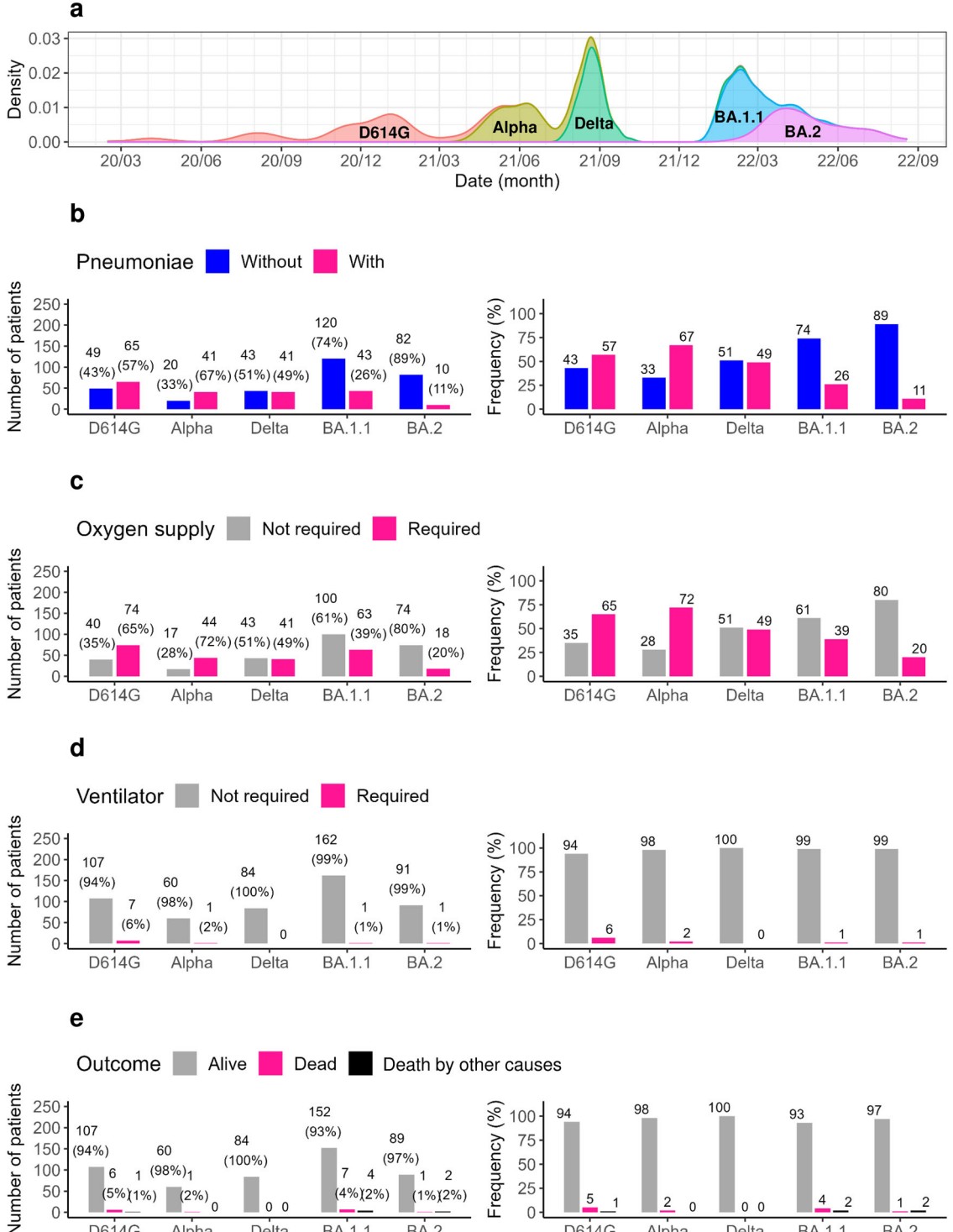

**Fig. 1 Clinical characteristics by variant. a** Historical transition of SARS-CoV-2 variants identified from February 2020 to August 2022. During this observation period, we identified the following numbers of hospitalized patients infected with the variants: D614G (*n* = 114), Alpha (*n* = 61), Delta (*n* = 84), BA.1.1 (*n* = 163), and BA.2 (*n* = 92). **b** Number of patients (left panel) and percentage of patients (right panel) that developed pneumonia by variant. **c** Number (left panel) and percentage (right panel) of patients requiring supplemental oxygen by variant. **d** Number (left panel) and percentage (right panel) of patients requiring ventilator use by variant. **e** Number (left panel) and percentage (right panel) of patient outcomes by variant. Numbers above bars indicate number of cases.

aspiration pneumonia) and was not considered to be a result of COVID-19 severity.

The number of patients requiring ventilator use also successively reduced as the variants changed: D614G (6%, 7/114), Alpha (2%, 1/61), Delta (0%, 0/84), Omicron BA.1.1 (1%, 1/163), and

Omicron BA.2 (1%, 1/92) (Fig. 1d). As the virus evolved, the number of patients requiring ventilator use approached zero. The following numbers of patients died after hospital admission: D614G (5%, 6/114), Alpha (2%, 1/61), Delta (0%, 0/84), Omicron BA.1.1 (4%, 7/163), and Omicron BA.2 (1%, 1/92) (Fig. 1e).

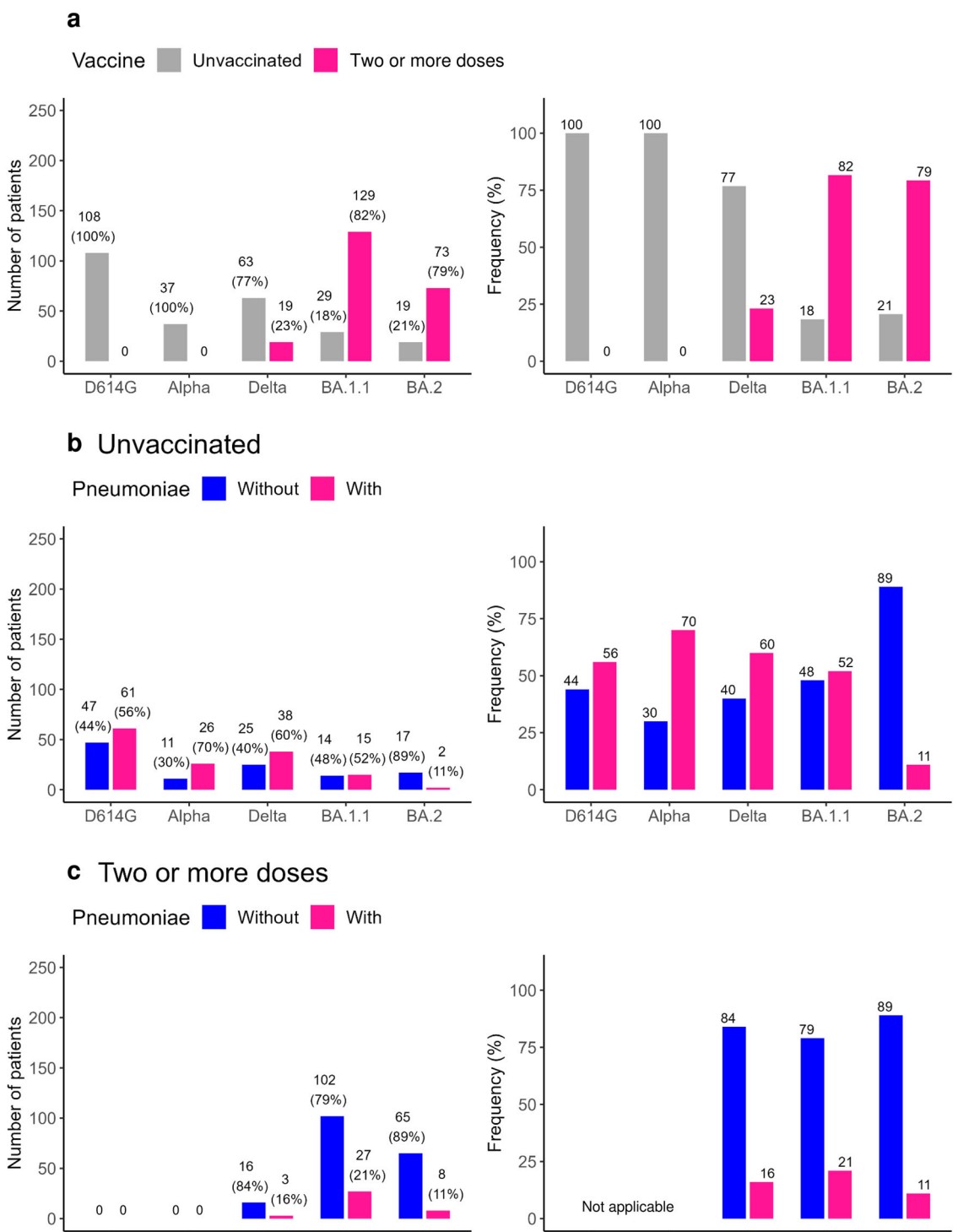

**Fig. 2 Vaccination status and pneumonia findings by variant. a** Vaccination status by variant. Number of patients (left panel) and percentage of patients (right panel) classified as unvaccinated or fully vaccinated (i.e., two or more doses). **b**, **c** Presence or absence of pneumonia by variant in unvaccinated (**b**) and fully vaccinated patients (**c**). Numbers above bars indicate number of cases.

**SARS-CoV-2 variant according to vaccination status and pneumonia findings**. We examined the variant according to vaccination status for the 477 individuals with a known vaccination history; this analysis excluded the 37 individuals with an unknown vaccination history. We observed the following numbers of variant infections among those who were fully vaccinated (i.e., those who had received at least two vaccine doses): D614G (0%, 0/108), Alpha (0%, 0/37), Delta (23%, 19/82), BA.1.1 (82%,

129/158), and BA.2 (79%, 73/92) (Fig. 2a). The number of people who were fully vaccinated increased sharply from 0% for those infected with D614G to 82% for those infected with BA.1.1 but was relatively stable at 79% for those infected with BA.2 (Fig. 2a).

Next, we examined the variant according to pneumonia findings and vaccination status. The number of unvaccinated patients with COVID-19-typical pneumonia remained almost unchanged across the variants (D614G (56%, 61/108), Alpha

(70%, 26/37), Delta (60%, 38/63), and BA.1.1 (52%, 15/29) (Fig. 2b). However, for those infected with the BA.2 genotype, fewer unvaccinated patients displayed pneumonia symptoms (11%, 2/19) (Fig. 2b).

We also examined the presence of pneumonia in fully vaccinated patients. None of the patients infected with either D614G or Alpha were fully vaccinated; therefore, these patients were not assessed further. COVID-19-typical pneumonia was apparent in 16% (3/19) and 21% (27/129) of vaccinated patients infected with Delta and BA.1.1, respectively; this is a 1/3 to 1/4 decrease compared with unvaccinated patients (60% and 52%) (Fig. 2c). In contrast, vaccinated and unvaccinated BA.2-infected patients showed similar rates of COVID-19-typical pneumonia by CT imaging (vaccinated: 11%, 8/73; unvaccinated: 11%, 2/19) (Fig. 2c).

**SARS-CoV-2 variants and lung involvement**. A quantitative CT score was calculated to evaluate the changes in pulmonary involvement in patients infected with the different variants[29]. The median total CT score in patients infected with D614G was 5 (IQR: 0.25–9), Alpha was 10 (IQR: 2–14), Delta was 2 (IQR: 0–11), BA.1.1 was 2 (IQR: 0–7), and BA.2 was 1 (IQR: 0–4) (Fig. 3). Compared with D614G, the total CT scores of patients with Alpha were significantly higher ($P = 0.031$), whereas those with BA.1.1 ($P = 0.031$) and BA.2 ($P = 7.0 \times 10^{-5}$) were significantly lower (Fig. 3, $P$-values were adjusted for multiple comparisons). Except for the right middle lobe, the CT scores showed similar trends in the other lung lobes according to the SARS-CoV-2 variant (Supplementary Fig. 1). These results suggest that an Omicron infection does not spread as readily into the lungs.

**Breakthrough infection in vaccinated BA.2-infected patients with pneumonia**. We next analyzed the BA.2-infected breakthrough infection cases in detail. Eight BA.2-infected patients developed COVID-19-typical pneumonia according to the CT

images (Fig. 4 and Supplementary Data 2). Case #8 showed atypical consolidation because the left lower lung was atelectatic. However, based on the clinical course of the disease, we diagnosed the patient with COVID-19 typical pneumonia because the right lung showed ground-glass shadows. The median age of these patients was 79 years (range 68–90 years), with three females and five males. Comorbidities included interstitial pneumonia ($n = 1$), hyperlipidemia ($n = 1$), diabetes ($n = 2$), hypertension ($n = 2$), and cancer ($n = 1$) (Supplementary Data 2). There were no secondary complications such as lung embolism, pneumothorax, or pneumoperitoneum. Of these patients, seven survived and one passed away.

One patient had received two vaccine doses, and seven had received three vaccine doses. All patients were vaccinated with BNT162b2 (Supplementary Data 2). The median duration from the time of last vaccination to infection was 6.6 weeks (range: 2.4–26.7 weeks). The anti-N antibody titers were negative in all cases, with a COI range of 0.07–0.17, implying that these patients had no previous history of infection. The pneumonia distributions were central ($n = 2$), peripheral ($n = 1$), and peripheral and central ($n = 5$), and the main patterns were CON ($n = 2$), CPA ($n = 2$), and GGO ($n = 4$).

We investigated whether the time between the most recent vaccination and infection was associated with disease progression. The anti-S antibody titers decreased with time since the most recent vaccination ($R = -0.94$, $P = 4.6 \times 10^{-4}$) (Supplementary Fig. 2a), but no correlation was found between this time period and the CT score ($R = 0.23$, $P = 0.59$) (Supplementary Fig. 2b). In addition, although the number of cases was small, the time between vaccination and infection was not associated with the degree of symptoms or the outcome (Supplementary Fig. 2c and 2d, Mann–Whitney $U$-test).

**Pneumonia findings in unvaccinated BA.2-infected patients**. The median age of the 19 unvaccinated BA.2-infected patients was 76 years (range 8–92 years), with 12 females and 7 males (Fig. 5). Four of these patients required supplemental oxygen. One of these patients required the use of a ventilator, and all of these patients survived. The median anti-S antibody titer was 0.4 U/mL (range 0.4–9476 U/mL, $n = 16$). The anti-N antibody titers ranged from 0.07 to 88.5 COI ($n = 17$). Five patients (cases #15, #17, #20, #21, and #26) had elevated S antibody titers. Case #15, #20, #21, and #26 were considered to have an immune response because sufficient time had passed since the infection. Case #17 had been previously infected with SARS-CoV-2. Of these 19 patients, 10 had no signs of pneumonia, seven showed COVID-19-atypical pneumonia, and only two showed COVID-19-typical pneumonia (Fig. 5 and Supplementary Data 2).

**Comparison of unvaccinated and fully vaccinated BA.2-infected patients who had COVID-19-typical pneumonia**. In this study, of the ten BA.2-infected patients with COVID-19-typical pneumonia, two were unvaccinated (11%, 2/19) and eight were fully vaccinated (11%, 8/73). Although the vaccination rate was 79% among those infected with BA.2 (Fig. 2a), the number of patients with findings of COVID-19 typical pneumonia remained high, and one patient died. One unvaccinated patient was treated with baricitinib, but not with antiviral or antibody therapy (Supplementary Data 2). Of the eight fully vaccinated patients, five were treated with remdesivir, one with remdesivir + baricitinib + dexamethasone, one with remdesivir + dexamethasone, and one with remdesivir + sotrovimab (Supplementary Data 2). Although not significant, the median total CT score tended to be lower in fully vaccinated patients compared with unvaccinated patients ($P = 0.066$, Mann–Whitney $U$-test)

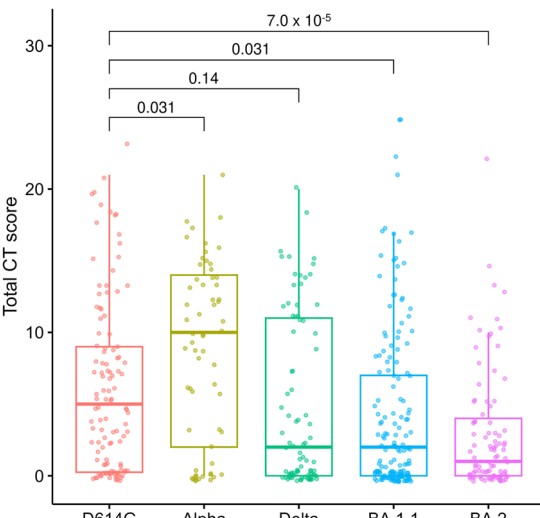

**Fig. 3 Relationship between SARS-CoV-2 variant and CT score.** The CT scores were evaluated semi-quantitatively based on the percentage of lung involvement in the five lung lobes, and the sum of the CT scores (total CT score) was calculated. Box plots show the total CT scores of D614G ($n = 114$), Alpha ($n = 61$), Delta ($n = 84$), BA.1.1 ($n = 163$), and BA.2. ($n = 92$). Each box indicates the interquartile range (top: the third quartile; bottom: the first quartile) with a horizontal line indicating the median. Statistical analysis was performed by a $t$-test, and $P$-values were adjusted for multiple comparisons.

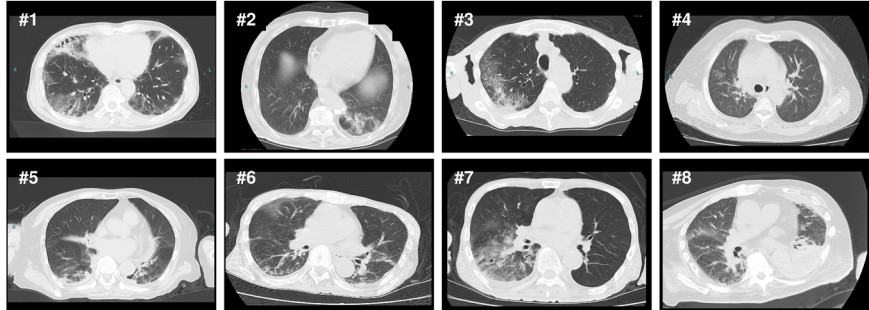

**Fig. 4 Eight fully vaccinated patients showing signs of pneumonia after BA.2 infection.** Chest CT images of the eight fully vaccinated cases who shows COVID-19 pneumoniae. These CT scans were taken immediately upon admission to the hospital.

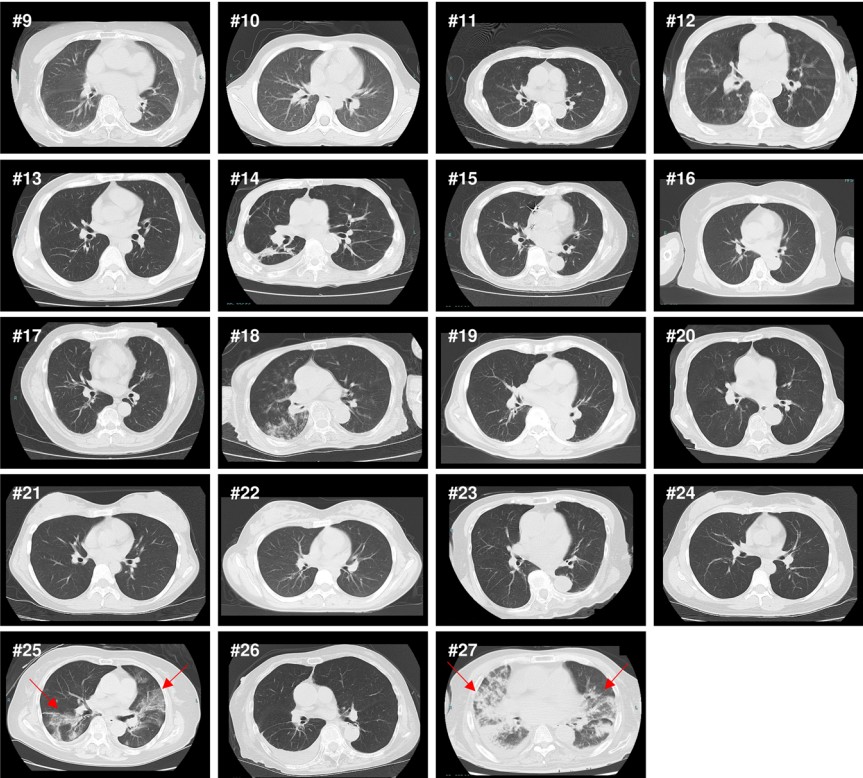

**Fig. 5 Nineteen cases of BA.2 infection in unvaccinated patients.** Chest CT images of the 19 unvaccinated cases who infected with BA.2. These CT scans were taken immediately upon admission to the hospital. Red arrows indicate sites of COVID-19 pneumonia.

(Supplementary Fig. 3). These results suggest that while the vaccine may be effective against the spread of lung lesions, frail and elderly individuals with some comorbidities infected with BA.2 are at risk of developing pneumonia and mortality[30], suggesting limitations of vaccines designed based on the original strain.

## Discussion

In this study, hospitalized patients underwent CT imaging to assess their pneumonia status. Consecutive data acquisition from the early to the most recent SARS-CoV-2 variant outbreaks showed that a higher proportion of hospitalized patients with pneumonia were male, older, had poorer outcomes, and were unvaccinated. Furthermore, the proportion of hospitalized patients with pneumonia showed a sharp downward trend with the successive changes in SARS-CoV-2 variant. The percentage of hospitalized patients with COVID-19-typical pneumonia peaked at 67% for the Alpha variant and then decreased markedly to 49%

for Delta, 26% for BA.1.1, and 11% for BA.2, marking a major decline in pneumonia prevalence from Delta to Omicron. In particular, there was a prominent decrease in pneumonia cases for those infected with BA.2.

Studies in animal models have reported that compared with Delta, the Omicron variants are more infectious in the upper respiratory tract but are less likely to invade the lungs and are less virulent[31,32]. Consistent with these trends, our results showed a lower rate of pneumonia in hospitalized patients infected with the Omicron variants than with the other SARS-CoV-2 variants (Supplementary Data 1). CT findings showed an organizing pneumonia pattern was typical in the early phase of the pandemic, but a bronchiolitis pattern was more common after the emergence of Omicron. Both the incidence of pneumonia and CT score were lower in Omicron-infected patients, suggesting Omicron is less aggressiveness. This implies a potential scenario in which viral evolution has led to increasing infectivity but decreasing virulence[33,34]. Nevertheless, old age was identified as a

risk factor for developing pneumonia in patients infected with the Omicron after receiving an inactivated vaccine injection[35]. This data is supported in the present study, which shows that older, high-risk patients with comorbidities who are infected with the Omicron BA.2 variants are still at risk of developing pneumonia and need early treatment to avoid severe disease.

When considering changes in the virulence of SARS-CoV-2, it is crucial to determine whether (1) tropism has changed and infection of the lung tissue has decreased, (2) the changes are due to the effect of vaccination, and (3) whether both of these factors are involved. In particular, compared with the previous variants, the Omicron variants contain more mutations, are more infectious, and have greater immune evasion[36]. Vaccine effectiveness increases with the third dose[37], but the overall vaccine effectiveness has been attenuated over time as the virus has evolved from the Delta variants to the Omicron variants[38]. Notably, of the eight patients who had a breakthrough infection with BA.2, six developed COVID-19-typical pneumonia despite having a high anti-S antibody titer and a short duration since their third vaccine dose (2.4–10.9 weeks). The anti-S antibody titers may have been raised by the booster dose, but the extent to which these antibodies have neutralizing activity in hospitalized patients needs to be evaluated. The first developed vaccines were designed against the original virus and were highly effective in the early phase of pandemics. However, it should be noted that conventional vaccines may not adequately protect against infection or prevent severe disease because Omicron has numerous mutations in the *S* gene.

Recent studies have shown that the Omicron variants have reduced virulence resulting from alterations in tropism. The Omicron variants have been reported to utilize cathepsin-dependent endosomal entry more than TMPRSS2-dependent plasma membrane entry[32,39,40]. Furthermore, Omicron replicates faster in the bronchi than the other SARS-CoV-2 variants, but it is less efficient at infecting the lung parenchyma[41]. Experimental infection in hamsters showed that BA.2 is more likely than BA.1 to infect lung tissue[42], but this was not supported by our clinical data. In the present study, the symptoms of the BA.2-infected patients were due to laryngitis and bronchiolitis, and the patients with BA.2 infection rarely showed clinical conditions consistent with pulmonary parenchymal damage. We speculate that BA.2 has decreased tropism for infection of the alveolar epithelium and may be more confined to the bronchiolar epithelium. Our study showed that lung involvement was significantly less common in BA.2-infected patients, which was in line with the CT scoring evaluation results. In other words, owing to the change in tropism, the pathogenicity of COVID-19 has changed so that it is less likely to result in lung parenchymal damage.

This study had several limitations. First, this dataset included hospitalized patients from only a single institution. Second, there were many confounding factors such as age, vaccination status, the time between the most recent vaccination and infection, and public health measures that complicated examining the relationship between the variants and the presence or absence of pneumonia. Third, although CT scans were performed immediately after admission, they were not performed subsequently; thus, the lung images could not be evaluated over time. Therefore, to accurately assess when COVID-19 pneumonia first appeared and when the condition worsened was not possible. Fourth, there was insufficient information on the history of prior infections for all patients. Ideally, separating the truly immunologically naïve patients from the previously infected patients who had infection-elicited immunity in the unvaccinated group would be preferable. To obtain some information about this, we evaluated the seropositivity of anti-N antibodies as a surrogate indicator of previous infection in the unvaccinated patients. Of the 256 unvaccinated patients, 48 (18.8%) were seropositive for N antibodies at admission, 205 (80.1%) were seronegative, and 3 (1.2%) were untested. Thus, it can be inferred that almost of the unvaccinated patients in our cohort had no history of previous infection.

In summary, the analysis of this dataset obtained over time showed that the proportion of COVID-19 patients with pneumonia decreased as the SARS-CoV-2 variant changed. Using CT images as a surrogate indicator of viral proliferation in the lung parenchyma, and hence pneumonia induction, the results suggest that BA.2 tropism in the lungs is decreased, resulting in a shift to upper respiratory tract proliferation rather than a pneumonia-inducing type of infection. Studies have suggested that governments gradually lessen the rigor of infection control measures and the use of non-pharmaceutical interventions when viral virulence decreases and herd immunity is achieved owing to an increased number of vaccinated individuals and previously infected individuals in the general population[43,44]. Over time after vaccination, serum antibody levels and vaccine effectiveness decrease, so additional vaccinations may be necessary depending on the disease severity caused by the variant circulating in the community[45]. In the future, additional vaccination may be promoted preferentially in the older adult population and for those at high risk of disease progression owing to underlying diseases[46,47]. To be taken into consideration is the fact that the vaccines designed for original virus are already not compatible with the currently circulating Omicron variants because of a poor antigenic match. In that context, the Omicron BA.4/BA.5-adapted bivalent COVID-19 vaccines are expected to protect against Omicron subvariants and to have improved efficacy against severe disease[48]. In addition, if intranasal vaccinations and nasal sprays become more developed, suppressing viral growth and clearing viruses in the nose and the upper respiratory system by triggering mucosal immune responses would deter viral spread[49–51]. To improve vaccine efficacy, the SARS-CoV-2 vaccine may need to be updated similar to the annual influenza vaccine.

## Data availability

The sequences of SARS-CoV-2 genomes are available on GISAID (www.gisaid.org). The source data for Figs. 1–3, Table 1, Supplementary Figs. 1-3, Supplementary Tables 1, 2, and Supplementary Data 1, 2 are in Supplementary Data 3.

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

## Acknowledgements

The authors would like to thank the clinical laboratory technicians in the microbiology laboratories at our institution. We also thank Katherine Thieltges from Edanz (https://jp.edanz.com/ac) for editing a draft of this manuscript. This study was supported by a Grant-in-Aid for the Genome Research Project from Yamanashi Prefecture (to M.O. and Y.H.), the Japan Society for the Promotion of Science (JSPS) KAKENHI Early-Career Scientists JP18K16292 (to Y.H.), a Grant-in-Aid for Scientific Research (B) 20H03668 (to Y.H.), a Research Grant for Young Scholars (to Y.H.), the YASUDA Medical Foundation (to Y.H.), the Uehara Memorial Foundation (to Y.H.) Medical Research Grants from the Takeda Science Foundation (to Y.H.), and Kato Memorial Bioscience Foundation (to Y.H.).

## Author contributions

Y.H. drafted the manuscript, generated the genomic data, and performed the statistical analyses. Y.K. and A.S. collected and interpreted the clinical data. T.T., S.H., H.Y., S.I., M.K., and H.K. participated in the study and interpreted the clinical data. Y.M. collected data and supervised the study. M.O. conceptualized the study design and revised the manuscript. All authors reviewed and approved the manuscript.

## Competing interests

The authors declare no competing interests.
