## [Peer Review File · Communications Medicine]

Reviewers' comments:

Reviewer #1 (Remarks to the Author):

Many thanks for the opportunity to Review this manuscript entitled: SARS-CoV-2 genotype and reduced lung tropism in hospitalized patients: A longitudinal study from D614G to Omicron BA.2.

I find the manuscript very interesting, however some major adjustments are required in order to increase and improve its quality.

- In the summary, report the right number of patients with pneumonia because the sums of patients with pneumonia is 194 and not 195, as authors reported in the results section in the text.

- Authors should report Sars.Cov-2 infections in the fully vaccinates as breakthrough infections and should define them in the introduction and report and discuss some cases of breakthrough infections with pneumonia with a little review of the literature in the discussion.

- Authors should define cases of patients that were fully vaccinated (more than 14 days from the second dose, or from the first and second booster) as breakthrough infections with pneumonia also in the methods. Authors should make more clear in a Table the group of the breakthrough infections with pneumonia, specifying the different variants and reporting also the time from infection and complete vaccination. Authors should also specify what vaccine was made (Pfizer, Moderna..), and should better summarize what comorbidities affected the patients and the age.

- Authors should specify in the same Table the age and comorbidities also for the unvaccinated groups.. Therefore authors should specify the CT-severity score for COVID-19 pneumonia in all these groups (score of Pan et al 2020, or Chung et al 2020) and if there are typical or atypical pattern of COVID-19 pneumonia (Typical pattern : peripheral or peripheral or central distribution). Authors should also report if there were cases of secondary complications as lung embolism or pneumothorax and pneumomediastinum.

- In the Figure 3 and in the Figure 4 authors should specify in the COVID-19 pattern if the distribution is typical or atypical for COVID-19 pneumonia and not with bronchial pattern..therefore, authors should edit the section of the pneumonia pattern. Authors should specify also if there are consolidation, GGO...

- Authors should also specify if other therapies have been made as monoclonal antibody, antiviral therapies, in patients with breakthrough infections with pneumonia.

- Authors should improve the discussion and extend the conclusion. In the conclusion authors reported: Therefore, the need for additional vaccination against BA.2 must be thoroughly discussed. I agree with the authors and they should also discuss this issue. There are a lot of articles that are reporting a reduction of the immune response after the booster dose of Mrna vaccines. Please report and discuss these studies:

Seneff, S., Nigh, G., Kyriakopoulos, A. M., & McCullough, P. A. (2022). Innate immune suppression by SARS-CoV-2 mRNA vaccinations: The role of G-quadruplexes, exosomes, and MicroRNAs. *Food and Chemical Toxicology*, 164, 113008.

Azim Majumder, M. A., & Razzaque, M. S. (2022). Repeated vaccination and 'vaccine exhaustion': relevance to the COVID-19 crisis. *Expert Review of Vaccines*, 1-4.

Yamamoto, K. (2022). Adverse effects of COVID-19 vaccines and measures to prevent them. *Virology Journal*, 19(1), 1-3.

Therefore, authors should modify and summarize better the conclusion also in the abstract. Can authors propose other alternative approaches: as spray nasal vaccines..or also improvement of Sars-Cov-2 vaccinations with vaccines that are less influenced from virus variants.. On the other hand, a current problem of the actual vaccines is that they don't block the virus transmission...and also early therapy could be necessary in patients with comorbidities to avoid severe disease also if they were fully vaccinated. Please discuss better these issues

Reviewer #2 (Remarks to the Author):

I had the pleasure to review the manuscript entitled "SARS-CoV-2 genotype and reduced lung tropism in hospitalized patients: A longitudinal study from D614G to Omicron BA.2". Although the topic is interesting, the information provided by the manuscript are not novel and already known.

The methodology of the study is not perfect. The criteria of the diagnosis of COVID-19 pneumonia on CT is not ideal. Although the authors considered peripherally distributed bilateral and multilobar ground-glass opacities, as characteristic CT findings of the disease, these findings are not very sensitive (although typical).

Not enough information about the imaging technique, time to image from the onset of symptoms, experience of radiologist reading the exam, etc has been provided.

The sample size is small.

I cannot recommend this manuscript for publication in Communications Medicine

Reviewer #3 (Remarks to the Author):

This article, entitled "SARS-CoV-2 genotype and reduced lung tropism in hospitalized patients: A longitudinal study from D614G to Omicron BA.2", examined the relationships between the genotype evolution of SARS-CoV-2 in 2020-2021 and its lung involvement which were diagnosed by computed tomography. The authors concluded that reduced lung tropism was noted in the latest surge caused by Omicron BA.1 and BA.2.

Firstly, this is a manuscript written with fluent and smart English, and the logic thinking was clear. Secondly, while the case number was not large enough, and the cases were from a single study site, the authors still made a very stringent inference, and their viewpoints were persuasive.

Nonetheless, some points deserved clarification by the authors.

Summary

1. At line 33-35, the authors' conclusion "The importance of additional COVID-19 vaccine doses should be reconsidered." was not straightforward, and made the readers confused.

Please state the Japanese policy of additional COVID-19 vaccine first, then make a reappraisal.

Introduction

2. In 1st paragraphs, please cite references at line 44-45 and line 48-49.

Methods

3. When did CT scan be performed? At the beginning of admission, during progression, or before discharge?

4. The contents of standardized medical record reviewed were not shown.

Results

5. At line 141, the meaning of “have different genotypes ($p < 0.001$)” was not clear.

6. All of 10 unvaccinated patients infected with BA.2 did not developed pneumonia, according to the 2nd paragraph of subheading Viral Genotype According to Vaccination Status and Pneumonia Findings at line 202-204. However, Figure 4 (A) still showed 4 cases with pneumonia. Please make a verification.

Discussions

7. At line 270, “this hypothesis is supported by the CT images of the large number of cases included in this study”. In this study, only 41 cases of BA.2 infection were included. The case number was not large. And, the authors may cite references to prove the change of tropism of BA.2 variant.

8. At line 276-278, “the present study did not examine vaccine efficacy in preventing infection, particularly with regard to differences in genotype.’ The authors stated. However, this study do examine the vaccine efficacy in preventing severe diseases, 16% and 21% of vaccinated patients acquiring delta and BA.1.1 infection respectively, which were 1/3 to 1/4 decrease compared with unvaccinated patients (60% and 56%). Therefore, the authors should propose the rationale that the unvaccinated patients did not develop pneumonia while acquiring BA.2 infection.

9. At line 285, the authors depicted that “the presence of circulating antibodies might not have had any therapeutic effect” may not be true. Until now, most people were japed COVID-19 vaccines that were designed from wild type of Wuhan. Meanwhile, monoclonal antibodies, like casirivimab + imdevimab, bamlanivimab + etesevimab, or sortrovimab, were not designed specifically against BA.2. Thus, it’s no doubt both present vaccine-mounted antibodies and infusion of current monoclonal antibodies were not effective against BA.2 infection. However, it may not be true when tixagevimab + cilgavimab become wildly used.

10. At line 289, the study limitations revealed the use of antibody and/or antiviral therapy may be confounders of the relationships between viral variants and presence of absence of pneumonia. Please show the use of antibody and antiviral agents in Results, and interpret the impacts.

11. At line 299, there is an incomplete sentence:and the spread of infection in the general population,

12. At line 308-309, the authors concluded that “the need for additional vaccination against BA.2 must be thoroughly discussed.”. First, the authors could make a reappraisal to Japanese policy of additional COVID-19 vaccine. Second, next generation COVID-19 vaccines could have been available when this article could be seen in public. Therefore, the conclusion may not be suitable in the near future.

Reviewer #4 (Remarks to the Author):

Hirotsu et al. examined the prevalence of COVID-19 pneumonia across different SARS-CoV-2 variant waves in unvaccinated and vaccinated hospitalized patients. The authors found that pneumonia prevalence declined with newer variants, possibly indicating a change in viral tropism to favour the upper respiratory tract. The authors also found that patients with COVID pneumonia are more likely to be unvaccinated, older, men, and had poorer outcomes (death). Interestingly, for BA.2 infection, there were 5 cases of vaccinated individuals who had pneumonia while none of the unvaccinated individuals developed pneumonia, in contrast to Delta and BA1.1, which had higher prevalence of pneumonia in unvaccinated individuals.

This study has a impressive and important dataset measuring the prevalence of COVID pneumonia spanning 2.5 years and 5 major SARS-CoV-2 variants. However, the manuscript could benefit from some further analysis as suggested below. The conclusion that vaccination should be reconsidered for BA.2 infection is unwarranted based on the limited sample size of unvaccinated BA.2 infected individuals.

Major comments:

1. For Fig 2B: Did any of the unvaccinated patients have history of prior infection? They should be separated from the truly immunologically naïve unvaccinated group as they would have some infection-elicited immunity despite being unvaccinated.
2. Line 141: what do the authors mean by “have different genotypes”? Have different viral lineages that are not ancestral D614G?
3. Table 1: p valueb: Wilcoxon rank–sum test; Pearson's chi-square test; Fisher's exact test
Authors should specify which test was used for each of the p values listed.
4. Fig 2B/2C, Were there any differences between the vaccinated people who got pneumonia vs those who didn't? (Age? Sex? N or S Ab titres? Time since last vaccine?)
5. Do the authors have any data on when symptoms of pneumonia started showing post-symptom onset or post-positive test? And has this changed over time with the new variants? (i.e., has disease progression become quicker/slower or remained unchanged?)
6. Fig 3: Time since last vaccination can be a factor in determining disease progression/outcomes following infection (the longer since last vaccination, the lower the residual immunity). Do the authors have this data for the five BA2 infected vaccinated individuals in Fig 3?
7. Could the authors add arrows to the images in Fig 3 to show signs of COVID19 pneumonia (peripherally distributed bilateral and multilobar ground glass opacities)? And describe how this differs from the bronchial pneumonia described in Fig 4 for cases #6, #9, #11 and #14, that wasn't classified as COVID19 pneumonia?
8. While it is interesting that none of the unvaccinated BA.2 infected patients showed signs of

COVID pneumonia, the sample size of unvaccinated BA.2 infected patients is rather small at n=10 compared to n=31 for vaccinated BA.2 infected patients, or compared to the BA1.1 infected groups (n=27 unvaccinated, n=126 vaccinated). Have the authors been able to recruit more BA.1-infected unvaccinated individuals and have any of them had COVID pneumonia?

9. Were there any other differences between the unvaccinated and vaccinated BA.2 infected patients who had pneumonia? Did they receive the same treatment upon hospital admission? Did pneumonia take longer to develop in vaccinated individuals?

10. Lines 306-309 (and last sentence of abstract), it is not reasonable for the authors to conclude that additional vaccination to protect against BA.2 should be reconsidered. While it is curious that there is a higher incidence of COVID pneumonia due to BA.2 in vaccinated vs unvaccinated individuals, the small sample size of BA.2-infected unvaccinated individuals (n=10; including 1 patient with previous infection) makes it hard to draw this conclusion. There is clear evidence that booster vaccinations are effective at reducing symptomatic infection and hospitalisation against Omicron variants including BA.2:

10.1056/NEJMc2210093, 10.1016/S1473-3099(22)00309-7, 10.15585/mmwr.mm7139a2

Minor comments:

1. Fig 1A legend and y-axis should specify whether this is number of infections or number of hospitalised patients.

2. Fig 2: label should say “2 or more doses” or “at least 2 doses” instead of “more than 2nd dose”.

3. Line 42: population is becoming increasingly immune, not immunized.

4. Line 216: sentence should be rephrased as “five patients infected with BA.2 showed signs of pneumonia on the CT images despite being vaccinated”

5. Fig 3 title should be more descriptive. E.g. “Five cases of fully vaccinated individuals showing signs of pneumonia after BA.2 infection”

6. I’m not sure if lines 274-279 in the discussion are necessary since the study was not designed to examine vaccine efficacy.

Responses to the comments of Reviewer #1

【Comment】

Many thanks for the opportunity to Review this manuscript entitled: SARS-CoV-2 genotype and reduced lung tropism in hospitalized patients: A longitudinal study from D614G to Omicron BA.2. I find the manuscript very interesting, however some major adjustments are required in order to increase and improve its quality.

【Response】

Thank you for your peer review. We appreciate your constructive comments.

【Comment】

- In the summary, report the right number of patients with pneumonia because the sums of patients with pneumonia is 194 and not 195, as authors reported in the results section in the text.

【Response】

Thank you for pointing out. We have included one person infected with the Gamma variant who showed pneumonia. This is misleading. As Gamma was detected only one patient, we excluded Gamma from subsequent analyses. It is noted in the Method section as follow: (page 7-8, line 97-99)

Only one patient with the Gamma variant was detected during this period, but this case was removed from subsequent analyses because of the small number of Gamma variant detections [13].

【Comment】

- Authors should report Sars.Cov-2 infections in the fully vaccinates as breakthrough infections and should define them in the introduction and report and discuss some cases of breakthrough infections with pneumonia with a little review of the literature in the discussion.

【Response】

In the introduction, we described about breakthrough infections with some references as follow:

(page 6, line 72-73)

Breakthrough infections can occur in individuals who have received more than two doses of vaccine [7-9].

Breakthrough infected patients were also examined in detail with respect to eight hospitalized patients infected with BA.2. We discussed these cases in revised manuscript (please see our

response below).

【Comment】

- Authors should define cases of patients that were fully vaccinated (more than 14 days from the second dose, or from the first and second booster) as breakthrough infections with pneumonia also in the methods. Authors should make more clear in a Table the group of the breakthrough infections with pneumonia, specifying the different variants and reporting also the time from infection and complete vaccination. Authors should also specify what vaccine was made (Pfizer, Moderna..), and should better summarize what comorbidities affected the patients and the age.

【Response】

Fully vaccinated and breakthrough infection was specified in Method as follows:

(page 8, line 106-109)

People who had received at least two vaccine doses with more than 14 days passing since the second dose were considered fully vaccinated. Breakthrough infection was defined as a positive SARS-CoV-2 PCR or antigen test result in a fully vaccinated individual.

For breakthrough infections, we examined in detail on the eight patients infected with BA.2. We added to the results the time from the date of last vaccination to infection, the type of vaccine, the number of vaccinations, antibody levels, the age of the patient and comorbidities as follow:

(page 18, line 281-292)

We next analyzed the BA.2-infected breakthrough infection cases in detail. Eight BA.2-infected patients developed COVID-19-typical pneumonia according to the CT images (Figure 4 and Supplemental Table 3). The median age of these patients was 79 years (range 68–90 years), with three women and five men. Comorbidities included interstitial pneumonia (n=1), hyperlipidemia (n=1), diabetes (n=2), hypertension (n=2), and cancer (n=1) (Supplemental Table 3). There were no secondary complications such as lung embolism, pneumothorax, or pneumoperitoneum. Of these patients, seven survived and one passed away.

One patient had received two vaccine doses, and seven had received three vaccine doses. All patients were vaccinated with BNT162b2 (Supplemental Table 3). The median duration from the time of last vaccination to infection was 6.6 weeks (range: 2.4–26.7 weeks) (Figure 4). The anti-N antibody titers were negative in all cases, with a COI range of 0.07–0.17, implying that these patients had no previous history of infection.

【Comment】

- Authors should specify in the same Table the age and comorbidities also for the unvaccinated groups. Therefore authors should specify the CT-severity score for COVID-19 pneumonia in all these groups (score of Pan et al 2020, or Chung et al 2020) and if there are typical or atypical pattern of COVID-19 pneumonia (Typical pattern : peripheral or peripheral or central distribution). Authors should also report if there were cases of secondary complications as lung embolism or pneumothorax and pneumomediastinum.

【Response】

We have created a Supplemental Table 3 that specifies age and comorbidities for the unvaccinated group vaccinated groups as well. We also listed the total CT score, typical or atypical COVID-19 pneumoniae, distribution, and main pattern in the tables in Figures 4 and Figure 5. In these patients, no patient developed secondary respiratory complications (i.e. lung embolism or pneumothorax and pneumomediastinum).

【Comment】

- In the Figure 3 and in the Figure 4 authors should specify in the COVID-19 pattern if the distribution is typical or atypical for COVID-19 pneumonia and not with bronchial pattern..therefore, authors should edit the section of the pneumonia pattern. Authors should specify also if there are consolidation, GGO...

【Response】

We edited the section of the pneumonia pattern to COVID-19 pneumonia and classified typical or atypical. In tables of Figure 4 and Figure 5, we also in the table distribution and main pattern (i.e. consolidations, GGO and crazy-paving appearance).

【Comment】

- Authors should also specify if other therapies have been made as monoclonal antibody, antiviral therapies, in patients with breakthrough infections with pneumonia.

【Response】

We examined patients with breakthrough infection treated with monoclonal antibody therapy and antiviral therapy and created a Supplemental Table 2. We described treatments information in revised manuscript as follow:

(page 13, line 200-203).

Of the 38 breakthrough infections with pneumonia, remdesivir was administered in 29 (76%), casirivimab/imdevimab in 1 (2.6%), sotrovimab in 2 (5.3%), and molnupiravir in 4 (11%) cases; nirmatrelvir/ritonavir were not used for treatment (Supplemental Table 2).

【Comment】

Authors should improve the discussion and extend the conclusion. In the conclusion authors reported: Therefore, the need for additional vaccination against BA.2 must be thoroughly discussed. I agree with the authors and they should also discuss this issue. There are a lot of articles that are reporting a reduction of the immune response after the booster dose of Mrna vaccines. Please report and discuss these studies:

Seneff, S., Nigh, G., Kyriakopoulos, A. M., & McCullough, P. A. (2022). Innate immune suppression by SARS-CoV-2 mRNA vaccinations: The role of G-quadruplexes, exosomes, and MicroRNAs. Food and Chemical Toxicology, 164, 113008.

Azim Majumder, M. A., & Razzaque, M. S. (2022). Repeated vaccination and ‘vaccine exhaustion’: relevance to the COVID-19 crisis. Expert Review of Vaccines, 1-4.

Yamamoto, K. (2022). Adverse effects of COVID-19 vaccines and measures to prevent them. Virology Journal, 19(1), 1-3.

Therefore, authors should modify and summarize better the conclusion also in the abstract. Can authors propose other alternative approaches: as spray nasal vaccines..or also improvement of Sars-Cov-2 vaccinations with vaccines that are less influenced from virus variants.. On the other hand, a current problem of the actual vaccines is that they don't block the virus transmission...and also early therapy could be necessary in patients with comorbidities to avoid severe disease also if they were fully vaccinated. Please discuss better these issues

【Response】

When we increased the number of BA.2 infected patients in the analysis, we found that some BA.2 vaccinated breakthrough infected patients also showed pneumonia. Although the number of patients was zero at the time of the initial submission, the increased number of analyses changed the results. Therefore, in accordance with the reviewer's comments, the conclusions in the abstract have been changed as follows:

(page 4, line 42-46)

Compared with previous variants, there was a marked decrease in pneumonia prevalence and lung involvement in patients infected with Omicron owing to decreased tropism in the lungs that hindered viral proliferation in the alveolar epithelial tissue. Nevertheless, older, high-risk patients with comorbidities who are infected with an Omicron variant can still develop pneumonia and require early treatment.

Regarding nasal vaccines and adopted vaccines compatible with the Omicron strain, we have clearly stated in the Discussion section as follow:

(page 25, line 406-413)

The presently used vaccines have been adapted to protect against Omicron BA.1 or the BA.4/BA.5 variants and are expected to have improved efficacy against severe disease [46]. In addition, if the development of intranasal vaccines advances, suppressing viral growth in the upper respiratory system by inducing an immune response within the nasal mucosa and the upper respiratory tract would help to deter viral spread [47, 48]. To improve vaccine efficacy, the SARS-CoV-2 vaccine may need to be updated similar to the annual influenza vaccine.

Despite the low virulence of the Omicron strain, it can still cause severe disease with respect to high-risk patients, so the need for early treatment is clearly stated in the Discussion section as follow:

(page 21, line 346-349)

Nevertheless, this study shows that older, high-risk patients with comorbidities who are infected with the Omicron BA.2 variants are still at risk of developing pneumonia and need early treatment to avoid severe disease.

Responses to the comments of Reviewer #2

【Comment】

The methodology of the study is not perfect. The criteria of the diagnosis of COVID-19 pneumonia on CT is not ideal. Although the authors considered peripherally distributed bilateral and multilobar ground-glass opacities, as characteristic CT findings of the disease, these findings are not very sensitive (although typical).

【Response】

Regarding CT imaging, the description of the methods section was corrected. Regarding the diagnostic criteria for pneumonia, we have added (1) quantitative CT scoring, (2) principal findings on chest CT, and (3) evaluation of the distribution of lung abnormalities. We described these issues in Method section as follow:

(page 11, line 153-166)

Definitions of radiological terms are based on the standard glossary of terms for chest imaging reported by the Fleischner Society [25]. On the basis of previous reports, a pneumonia diagnosis was made and the main findings on chest CT were classified into three categories: consolidation (CON), crazy-paving appearance (CPA), and ground-glass opacity (GGO) [26-28]. The distribution of lung abnormalities was divided into three

sections from the pulmonary hilum to the subpleural area and classified as those that were only subpleural (peripheral), those that extended near to the pulmonary hilum (central), and those that extended to the middle area between the peripheral and central areas. Semi-quantitative CT scores were evaluated for the right upper lobe, right middle lobe, right lower lobe, left upper lobe, and left lower lobe based on previous reports [29]. Briefly, the CT scores were calculated according to the lesion area as follows: 0, 0% involvement; 1, <5% involvement; 2, 5%–25% involvement; 3, 26%–50% involvement; 4, 50%–75% involvement; 5, >75% involvement. The total CT score was the sum of the individual scores, ranging from 0 (no involvement) to 25 (maximum involvement).

【Comment】

Not enough information about the imaging technique, time to image from the onset of symptoms, experience of radiologist reading the exam, etc has been provided.

【Response】

CT imaging was performed immediately after admission. The experience of radiologist is 25 years. These information about the imaging method, time from onset to imaging, and experience of the radiologist reading the images has been added to the Method section as follow:

(page 10, line 144-149)

CT Imaging

All patients underwent CT immediately after arriving at the hospital. CT images (64×0.5 mm or 32×1.0 mm-detector row/automatic exposure control/120 kVp) were obtained with the Aquilion 64 or the Aquilion 64CX system (Canon Medical, Tochigi, Japan). The 0.5 mm or 1.0 mm-slice thickness high-resolution computed tomography images were reconstructed using the FC51 reconstruct function.

(page 11, line 152-153)

All CT images were evaluated by an experienced radiologist specialized in thoracic radiology with 25 years of experience.

【Comment】

The sample size is small.

【Response】

Thank you for your remarks about the small sample size. What we could do was to increase the number of newly hospitalized BA.1.1 and BA.2 patients after submission of the manuscript. We increased the number of patients by 55 (from 460 to 514) and re-analyzed

the data.

【Comment】

I cannot recommend this manuscript for publication in Communications Medicine

【Response】

Thank you for your peer review and cooperations. We have responded to your comments and would appreciate it if you would evaluate our work.

Responses to the comments of Reviewer #3

【Comment】

This article, entitled "SARS-CoV-2 genotype and reduced lung tropism in hospitalized patients: A longitudinal study from D614G to Omicron BA.2", examined the relationships between the genotype evolution of SARS-CoV-2 in 2020-2021 and its lung involvement which were diagnosed by computed tomography. The authors concluded that reduced lung tropism was noted in the latest surge caused by Omicron BA.1 and BA.2.

Firstly, this is a manuscript written with fluent and smart English, and the logic thinking was clear. Secondly, while the case number was not large enough, and the cases were from a single study site, the authors still made a very stringent inference, and their viewpoints were persuasive.

Nonetheless, some points deserved clarification by the authors.

【Response】

Thank you for your peer review. Time has passed since the manuscript was submitted and the total number of admissions for BA.1 and BA.2 has increased by 55 patients (total admissions increased from 460 to 514 patients). We have added that data and re-analyzed the data.

【Comment】

Summary

1. At line 33-35, the authors' conclusion "The importance of additional COVID-19 vaccine doses should be reconsidered." was not straightforward, and made the readers confused. Please state the Japanese policy of additional COVID-19 vaccine first, then make a reappraisal.

【Response】

When we increased the number of cases, we found that some unvaccinated patients with BA.2 also showed pneumonia. As we also think this conclusion is misleading, we removed this sentence.

【Comment】

Introduction

2. In 1st paragraphs, please cite references at line 44-45 and line 48-49.

【Response】

We added references to the text.

【Comment】

Methods

3. When did CT scan be performed? At the beginning of admission, during progression, or before discharge?

【Response】

The CT scan was performed immediately after the patient arrived to the hospital. This point was clearly stated in Method as follows:

(page 10 line 145)

CT imaging

All patients underwent CT immediately after arriving at the hospital.

【Comment】

4. The contents of standardized medical record reviewed were not shown.

【Response】

There was an incorrect content. We did not collect data in a standardized format for all cases. Correctly, we retrospectively examined electronic medical records. The text has been corrected as follows:

(page 8, line 104-106)

Medical staff collected data on the medical condition, vaccination status, and disease course from patient interviews and retrospectively examined the electronic medical records.

【Comment】

Results

5. At line 141, the meaning of "have different genotypes ($p < 0.001$)" was not clear.

【Response】

I agree that the meaning of the sentence is difficult to understand. We corrected as follow.

(page 13, line 198-200)

Compared with D614G, Alpha, and Delta variants, patients infected with Omicron BA.1.1 or BA.2 variants had significantly lower rates of pneumonia (adjusted $P < 0.01$, pairwise Fisher's exact test) (Supplemental Table 1).

【Comment】

6. *All of 10 unvaccinated patients infected with BA.2 did not developed pneumonia, according to the 2nd paragraph of subheading Viral Genotype According to Vaccination Status and Pneumonia Findings at line 202-204. However, Figure 4 (A) still showed 4 cases with pneumonia. Please make a verification.*

【Response】

Thank you for pointing out the ambiguous description. Due to the increase in the number of cases, we have re-examined the CT images. Figure 2 shows the number of patients with "typical" COVID-19 pneumonia and does not count "atypical" COVID-19 pneumonia. As you pointed out, to avoid confusion, we have changed the Table in Figure 4 to include "COVID-19 pneumonia" and revised the classification to "Atypical" and "Typical" COVID-19 pneumonia.

【Comment】

Discussions

7. *At line 270, "this hypothesis is supported by the CT images of the large number of cases included in this study". In this study, only 41 cases of BA.2 infection were included. The case number was not large. And, the authors may cite references to prove the change of tropism of BA.2 variant.*

【Response】

The number of cases in BA.2 has increased from 41 to 92. As noted in the comments, the number of cases is still not sufficient, so this description (large...) has been deleted.

【Comment】

8. *At line 276-278, "the present study did not examine vaccine efficacy in preventing infection, particularly with regard to differences in genotype." The authors stated. However, this study do examine the vaccine efficacy in preventing severe diseases, 16% and 21% of vaccinated patients acquiring delta and BA.1.1 infection respectively, which were 1/3 to 1/4 decrease compared with unvaccinated patients (60% and 56%). Therefore, the authors should propose the rationale that the unvaccinated patients did not develop pneumonia while acquiring BA.2 infection.*

【Response】

Thank you for your constructive comments. We have increased the number of cases and reanalyzed the data. There were two case (11%) showing typical COVID-19 pneumonia after BA.2 infection among the unvaccinated patients. Perhaps the small number of cases led to an incorrect interpretation at the time of the initial submission. Therefore, we removed this statement.

【Comment】

9. *At line 285, the authors depicted that “the presence of circulating antibodies might not have had any therapeutic effect” may not be true. Until now, most people were japed COVID-19 vaccines that were designed from wild type of Wuhan. Meanwhile, monoclonal antibodies, like casirivimab + imdevimab, bamlanivimab + etesevimab, or sortrovimab, were not designed specifically against BA.2. Thus, it’s no doubt both present vaccine-mounted antibodies and infusion of current monoclonal antibodies were not effective against BA.2 infection. However, it may not be true when tixagevimab + cilgavimab become wildly used.*

【Response】

As stated in the reviewer's comments, it was inappropriate and will be removed.

【Comment】

10. *At line 289, the study limitations revealed the use of antibody and/or antiviral therapy may be confounders of the relationships between viral variants and presence of absence of pneumonia. Please show the use of antibody and antiviral agents in Results, and interpret the impacts.*

【Response】

Information on patient’s treatment has been added to the Supplemental Table 2. The CT scan was performed immediately after admission and before treatment was initiated. Presumably, if post-treatment CT scans were analyzed, the effect of treatment must also be considered as a confounding factor, but this study did not consider it because CT scan performed right after the hospitalization. Therefore, we have removed the description that treatment is a confounding factor for the development of pneumonia.

【Comment】

11. *At line 299, there is an incomplete sentence:and the spread of infection in the general population,*

【Response】

The description has been revised as follows:

(page 24, line 398-402)

Studies have suggested that governments gradually lessen the rigor of infection control measures and the use of non-pharmaceutical interventions when viral virulence decreases and herd immunity is achieved owing to an increased number of vaccinated individuals and previously infected individuals in the general population

【Comment】

12. At line 308-309, the authors concluded that “the need for additional vaccination against BA.2 must be thoroughly discussed.”. First, the authors could make a reappraisal to Japanese policy of additional COVID-19 vaccine. Second, next generation COVID-19 vaccines could have been available when this article could be seen in public. Therefore, the conclusion may not be suitable in the near future.

【Response】

I agree with the reviewer's comments. Currently, Japan has approved vaccination not only against the original SARS-CoV-2, but also against BA.1 and BA.4/BA.5 adapted vaccine. Antibody titers against the In this scenario, the conclusions of this study could be misleading, so the concluding sentence was revised as follows:

(page 25, line 406-413)

The presently used vaccines have been adapted to protect against Omicron BA.1 or the BA.4/BA.5 variants and are expected to have improved efficacy against severe disease [46]. In addition, if the development of intranasal vaccines advances, suppressing viral growth in the upper respiratory system by inducing an immune response within the nasal mucosa and the upper respiratory tract would help to deter viral spread [47, 48]. To improve vaccine efficacy, the SARS-CoV-2 vaccine may need to be updated similar to the annual influenza vaccine.

Responses to the comments of Reviewer #4

【Comment】

This study has a impressive and important dataset measuring the prevalence of COVID pneumonia spanning 2.5 years and 5 major SARS-CoV-2 variants. However, the manuscript could benefit from some further analysis as suggested below. The conclusion that vaccination should be reconsidered for BA.2 infection is unwarranted based on the limited sample size of unvaccinated BA.2 infected individuals.

【Response】

I agree with you that the sample size is too small. In revised manuscript, we have added data from BA.2 patients who were subsequently admitted to the hospital. As a result, some

of the unvaccinated BA.2 patients also developed pneumonia. Therefore, due to the misleading conclusion that vaccination should be reconsidered, we deleted it and rewrite the conclusion.

【Comment】

Major comments:

1. For Fig 2B: Did any of the unvaccinated patients have history of prior infection? They should be separated from the truly immunologically naïve unvaccinated group as they would have some infection-elicited immunity despite being unvaccinated.

【Response】

I completely agree with your comments. In this cohort, we did not investigate the history of previous infections with respect to all patients, and we do not have any data of infection history. Therefore, it is possible that some previously infected individuals were included. This is an important point in interpreting the data, which is why we have included it as a limitation in the Discussion section as below. What we can do is to take the presence or absence of anti-N antibodies measured on admission as a surrogate indicator of a history of prior infection. Of course, we are aware that this is not as an ideal surrogate indicator, since N antibodies may rise with the course of infection, and N antibodies may become negative after a long period of time since the infection. The results show that of the 256 unvaccinated persons, 48 (18.8%) were positive for N antibodies at the time of admission, 205 (80.1%) were negative, and 3 (1.2%) were unmeasured. Thus, it can be inferred that 80.1% of all unvaccinated patients had no history of infection.

(page 23-24, line 384-392)

Fourth, there was insufficient information on the history of prior infections for all patients. Ideally, separating the truly immunologically naïve patients from the previously infected patients who had infection-elicited immunity in the unvaccinated group would be preferable. To obtain some information about this, we evaluated the seropositivity of anti-N antibodies as a surrogate indicator of previous infection in the unvaccinated patients. Of the 256 unvaccinated patients, 48 (18.8%) were seropositive for N antibodies at admission, 205 (80.1%) were seronegative, and 3 (1.2%) were untested. Thus, it can be inferred that 80.1% of the unvaccinated patients in our cohort had no history of previous infection.

【Comment】

2. Line 141: what do the authors mean by “have different genotypes”? Have different viral lineages that are not ancestral D614G?

【Response】

I agree that the meaning of the sentence is difficult to understand. We corrected as follow.
(page 13, line 198-200)

Compared with D614G, Alpha, and Delta variants, patients infected with Omicron BA.1.1 or BA.2 variants had significantly lower rates of pneumonia (adjusted $P < 0.01$, pairwise Fisher's exact test) (Supplemental Table 1).

【Comment】

3. *Table 1: p valueb: Wilcoxon rank–sum test; Pearson's chi-square test; Fisher's exact test*
Authors should specify which test was used for each of the p values listed.

【Response】

We have made sure that each p-value shows the statistical method. We clarified the statistical method in the note of the Table 1 as follows:

Statistical analysis was performed with Wilcoxon rank sum test †; Pearson's Chi-squared test ‡; Fisher's exact test §

【Comment】

4. *Fig 2B/2C, Were there any differences between the vaccinated people who got pneumonia vs those who didn't? (Age? Sex? N or S Ab titres? Time since last vaccine?)*

【Response】

We examined for differences between vaccinated patients who developed pneumonia and those who did not. Age tended to be higher in Delta and BA.1.1-infected individuals who developed pneumonia. The anti-S antibody titers were lower in Delta-infected patients who developed pneumonia and higher in BA.1.1-infected patients, and The anti-N antibody titers were higher in BA.1.1-infected patients who developed pneumonia. It was difficult to perform an analysis on the period of time since the last vaccination because data on the date of vaccination for many patients were not available. The data are attached as "Reviewer only".

【Comment】

5. Do the authors have any data on when symptoms of pneumonia started showing post-symptom onset or post-positive test? And has this changed over time with the new variants? (i.e., has disease progression become quicker/slower or remained unchanged?)

【Response】

Thank you for your very valuable comments. I agree that all of your comments regarding the progression of pneumonia are important. Unfortunately, it was difficult to get a clear answer for when the pneumonia started to appear, because we evaluated almost all of the

cases with only one CT scan immediately after admission. This constructive comment is important and has been added as a limitation of the study.

(page 23, line 381-384)

Third, although CT scans were performed immediately after admission, they were not performed subsequently; thus, the lung images could not be evaluated over time.

Therefore, to accurately assess when COVID-19 pneumonia first appeared and when the condition worsened was not possible.

【Comment】

6. Fig 3: Time since last vaccination can be a factor in determining disease progression/outcomes following infection (the longer since last vaccination, the lower the residual immunity). Do the authors have this data for the five BA2 infected vaccinated individuals in Fig 3?

【Response】

Additional cases were available, so data were examined for a total of eight BA.2-infected vaccinated patients. All were vaccinated with Pfizer's BNT162b2 (one with two doses and seven with three doses), and the median number of days since the last vaccination date was 6.6 weeks (range: 2.4–26.7 weeks). As noted in the reviewer's comments, anti-S antibody titers decreased as the time since vaccination increased. However, there were no significant differences in the severity of symptoms or outcome and the time from the date of vaccination to the onset of disease (possibly due to the small number of cases). We have added these data as Supplementary Figure 1. The following statement was added to Result section:

(page 18, line 289-292)

One patient had received two vaccine doses, and seven had received three vaccine doses. All patients were vaccinated with BNT162b2 (Supplemental Table 3). The median duration from the time of last vaccination to infection was 6.6 weeks (range: 2.4–26.7 weeks)

(page 19, line 296-302)

We investigated whether the time between the most recent vaccination and infection was associated with disease progression. The anti-S antibody titers decreased with time since the most recent vaccination ($R = -0.94$, $P < 0.001$) (Supplemental Figure 2a), but no correlation was found between this time period and the CT score ($R = 0.23$, $P = 0.59$) (Supplemental Figure 2b). In addition, although the number of cases was small, the time between vaccination and infection was not associated with the degree of symptoms or the outcome (Supplemental Figure 2c and 2d, Wilcoxon test).

【Comment】

7. *Could the authors add arrows to the images in Fig 3 to show signs of COVID19 pneumonia (peripherally distributed bilateral and multilobar ground glass opacities)? And describe how this differs from the bronchial pneumonia described in Fig 4 for cases #6, #9, #11 and #14, that wasn't classified as COVID19 pneumonia?*

【Response】

Case numbers have been changed due to an increase in the number of cases. In Case #25 and #27 of the images in Figure 5, the signs of COVID19 pneumonia are indicated by arrows on the CT images. And, thank you for pointing out the ambiguous description. Due to the increase in the number of cases, we have re-examined the CT images. Figure 4 shows the number of patients with "typical" COVID-19 pneumonia and does not count "atypical" COVID-19 pneumonia. As you pointed out, to avoid confusion, we have changed the Table in Figure 4 to include "COVID-19 pneumonia" and revised the classification to "Atypical" and "Typical" COVID-19 pneumonia.

【Comment】

8. *While it is interesting that none of the unvaccinated BA.2 infected patients showed signs of COVID pneumonia, the sample size of unvaccinated BA.2 infected patients is rather small at n=10 compared to n=31 for vaccinated BA.2 infected patients, or compared to the BA1.1 infected groups (n=27 unvaccinated, n=126 vaccinated). Have the authors been able to recruit more BA.1-infected unvaccinated individuals and have any of them had COVID pneumonia?*

【Response】

I completely agree with the reviewer's comment. After the manuscript was submitted, we reanalyzed the data by adding newly hospitalized patients; the number of BA.1.1 cases increased from 153 to 158, and the number of BA.2 cases increased from 41 to 92. The number of unvaccinated BA.2 cases increased to 19, including two cases showing pneumonia. Therefore, we will make changes to add data and revise the Result section.

【Comment】

9. *Were there any other differences between the unvaccinated and vaccinated BA.2 infected patients who had pneumonia? Did they receive the same treatment upon hospital admission? Did pneumonia take longer to develop in vaccinated individuals?*

【Response】

When we looked for differences between unvaccinated and vaccinated BA.2-infected patients who had pneumonia, we found that the CT score, which evaluates the spread of

pneumonia by CT imaging, was significantly higher in unvaccinated patients than in fully vaccinated patients (please see Supplemental Figure 3). They did not receive the same treatment at the time of hospitalization. Regarding the duration of onset to pneumonia, CT examinations were only performed at the time of admission, making continuous evaluation difficult. The details of what we were able to investigate in this study are clearly described in the Result section as follow:

(page 20, line 316-328)

Comparison of Unvaccinated and Fully Vaccinated BA.2-infected Patients who had COVID-19-typical Pneumonia

In this study, of the 10 BA.2-infected patients with COVID-19-typical pneumonia, two were unvaccinated and eight were fully vaccinated. One unvaccinated patient was treated with baricitinib, but not with antiviral or antibody therapy (Supplemental Table 3). Of the eight fully vaccinated patients, five were treated with remdesivir, one with remdesivir + baricitinib + dexamethasone, one with remdesivir + dexamethasone, and one with remdesivir + sotrovimab (Supplemental Table 3). Although not significant, the median total CT score tended to be lower in fully vaccinated patients compared with unvaccinated patients ($P = 0.066$, Wilcoxon test) (Supplemental Figure 3). Therefore, compared with fully vaccinated patients, the unvaccinated patients with BA.2 infection possibly had more severe pneumonia. Collectively, these results imply that the vaccine protected against the severity of COVID-19 pneumonia.

【Comment】

10. Lines 306-309 (and last sentence of abstract), it is not reasonable for the authors to conclude that additional vaccination to protect against BA.2 should be reconsidered. While it is curious that there is a higher incidence of COVID pneumonia due to BA.2 in vaccinated vs unvaccinated individuals, the small sample size of BA.2-infected unvaccinated individuals (n=10; including 1 patient with previous infection) makes it hard to draw this conclusion. There is clear evidence that booster vaccinations are effective at reducing symptomatic infection and hospitalisation against Omicron variants including BA.2: 10.1056/NEJMc2210093, 10.1016/S1473-3099(22)00309-7, 10.15585/mmwr.mm7139a2

【Response】

As you indicated, we increased the number of cases and included a patient with BA.2 infection who was unvaccinated and showed pneumonia. We reconsidered the last sentence is misleading and removed this description from the revised manuscript.

【Comment】

Minor comments:

1. *Fig 1A legend and y-axis should specify whether this is number of infections or number of hospitalised patients.*

【Response】

We have specified that this is a density plot of hospitalized patients. We revised in Figure legends as follow:

(page 28, line 437-438)

During this observation period, we identified the following numbers of hospitalized patients infected with the variants

【Comment】

2. *Fig 2: label should say “2 or more doses” or “at least 2 doses” instead of “more than 2nd dose”.*

【Response】

The label has been changed to "Two or more doses".

【Comment】

3. *Line 42: population is becoming increasingly immune, not immunized.*

【Response】

As you indicated, we have corrected it.

(page 6, line 64-66)

However, the population is becoming increasingly immune owing to an increase in the number of natural infections and increasing numbers of vaccinated individuals.

【Comment】

4. *Line 216: sentence should be rephrased as “five patients infected with BA.2 showed signs of pneumonia on the CT images despite being vaccinated”*

【Response】

Due to an increase in the number of cases, the number of patients has been changed from 5 to 8. We have changed the text as shown in the comments as follow:

(page 18, line 281-283)

Eight BA.2-infected patients developed COVID-19-typical pneumonia according to the CT images

【Comment】

5. *Fig 3 title should be more descriptive. E.g. "Five cases of fully vaccinated individuals showing signs of pneumonia after BA.2 infection"*

【Response】

We have changed the title as follow.

(page 32, line 463-464)

Eight fully vaccinated patients showing signs of pneumonia after BA.2 infection.

【Comment】

6. I'm not sure if lines 274-279 in the discussion are necessary since the study was not designed to examine vaccine efficacy.

【Response】

As you have indicated, the effects of the vaccine are not relevant to this paper, so this sentence was deleted.

Reviewers' comments:

Reviewer #1 (Remarks to the Author):

Many thanks for the major adjustments and also for the topic explored in this manuscript. Although breakthrough infections with COVID-19 pneumonia are common in clinical practice, there are few articles mainly case reports or case series that described the radiological features of the Breakthrough infections with COVID-19 pneumonia.

Authors should report in the table 1 the type of COVID-19 vaccines (MRNA PFIZER, MRNA MODERNA...) and should specify the time of the last dose because authors reported it only in the omicron variant section. COVID-19 vaccines were very effective in the early phases of the pandemic with the first variants. However, COVID-19 pneumonia was also possible in frail people and in older patients also if they were fully vaccinated:

Brogna, B., Bignardi, E., Brogna, C., Capasso, C., Gagliardi, G., Martino, A., & Musto, L. A. (2021). COVID-19 Pneumonia in Vaccinated Population: A Six Clinical and Radiological Case Series. *Medicina*, 57(9), 891.

With the omicron variant, authors reported 8 cases with the typical findings of COVID-19 pneumonia in fully vaccinated patients. However, the lobar consolidation in the case 8 of the vaccinated group is an atypical findings. Please edit this image with one that shows better the typical findings of COVID-19 pneumonia in this patient. Authors reported that the percentages of pneumonia is the same (8/73 in the vaccinated group and 2/19 in the unvaccinated). However, these findings means that 73 vaccinated patients were admitted to the Hospitals against the 19 not vaccinated. Therefore, in the section Comparison of Unvaccinated and Fully Vaccinated BA.2-infected Patients who had COVID-19-typical Pneumonia authors should report and enhance that most patients with BA.2 and typical finding of COVID-19 pneumonia were vaccinated and only one vaccinated died. Therefore, this result should be enhance in this section because it represents a limits of actual m RNA vaccines.

Could these features be explained by the reduction of the immune response that was hypothesized by some authors especially with multiple doses of COVID-19 m RNA vaccines?

Seneff, S., Nigh, G., Kyriakopoulos, A. M., & McCullough, P. A. (2022). Innate immune suppression by SARS-CoV-2 mRNA vaccinations: The role of G-quadruplexes, exosomes, and MicroRNAs. *Food and Chemical Toxicology*, 164, 113008.

I have already suggested this article but I don't find it in the discussion. Please discuss it

I suggest to discuss also this article in the discussion: Tong X, Huang Z, Zhang X, et al. Old Age is an Independent Risk Factor for Pneumonia Development in Patients with SARS-CoV-2 Omicron Variant Infection and a History of Inactivated Vaccine Injection. *Infect Drug Resist.* 2022;15:5567-5573.

doi:10.2147/IDR.S380005

Therefore, authors in the discussion should enhance also the limits of the actual COVID-19 vaccines that showed a high efficacy during the first phase of the pandemic. However, the short duration of the immune response after the vaccination and covid-19 pneumonia in the vaccinated patients were not reported in the clinical trials but we surprisingly found them in the clinical practice.

On the other hand, I think that with the omicron variant we can stop to describe the disease as COVID-19 because the disease is very different and less aggressive than the COVID-19 of the phirst phase of the pandemic . In fact, the unvaccinated group shows less frequent pneumonia than the vaccinated group and on imaging are more frequent atypical findings. Authors should describe better this finding in the discussion

These considerations are important because they would mean rethinking COVID-19 management policies.

I suggest also to report in the discussion the typical and atypical radiological findings of the

pneumonia of COVID-19 in the first phase of the pandemic.

Therefore, I suggest a minor revision

It is a wonderful work because it is original, contains clinical and radiological findings and reflect the clinical practice.

Congratulations for this work

Reviewer #2 (Remarks to the Author):

The authors have satisfactorily responded to the reviewer comments, but still the manuscript does not have a novel finding and message for our readers and does not contribute to the literature significantly

Reviewer #3 (Remarks to the Author):

In this revision, the authors had added case numbers of BA.2 infection, and the data became much more persuasive. However, at line 406-413 in Conclusion, the opinions regarding booster vaccination were not properly addressed and should be remodified. The authors may discuss the concerns of repeated vaccination which could possibly cause “vaccine exhaustion” (according to the comments of reviewer#1), the potential benefits of next generation bivalent COVID-19 vaccines, and, the novel nasal spray COVID-19 vaccines, which have possible impacts on both reducing disease severity and viral transmission.

In addition, the case numbers in figure legends of fig 4 (a) (b) and fig 5 (a) (b) were incorrect.

Reviewer #4 (Remarks to the Author):

The authors have addressed all my comments and concerns sufficiently.

One minor comment:

The authors should use the unpaired Mann-Whitney test for their comparisons in Supplemental Figures 2C, 2D and 3 instead of the Wilcoxon test since the data are unpaired. I don't think it'll change the results but the Wilcoxon test is inappropriate here.

----- Responses to the comments of Reviewer #1 -----

[Comment]

Many thanks for the major adjustments and also for the topic explored in this manuscript. Although breakthrough infections with COVID-19 pneumonia are common in clinical practice, there are few articles mainly case reports or case series that described the radiological features of the Breakthrough infections with COVID-19 pneumonia.

Authors should report in the table 1 the type of COVID-19 vaccines (MRNA PFIZER, MRNA MODERNA...) and should specify the time of the last dose because authors reported it only in the omicron variant section. COVID-19 vaccines were very effective in the early phases of the pandemic with the first variants. However, COVID-19 pneumonia was also possible in frail people and in older patients also if they were fully vaccinated:

Brogna, B., Bignardi, E., Brogna, C., Capasso, C., Gagliardi, G., Martino, A., & Musto, L. A. (2021). COVID-19 Pneumonia in Vaccinated Population: A Six Clinical and Radiological Case Series. *Medicina*, 57(9), 891.

[Response]

Thank you for your peer review. We looked up the vaccine type (Pfizer, Moderna) and the time from the date of the last vaccine administration to the infection from the electronic medical record and specified them in Table 1. We also cite the paper (Brogna et al 2021) in the Result section in page 20 line 330 (reference #30).

[Comment]

With the omicron variant, authors reported 8 cases with the typical findings of COVID-19 pneumonia in fully vaccinated patients. However, the lobar consolidation in the case 8 of the vaccinated group is an atypical findings. Please edit this image with one that shows better the typical findings of COVID-19 pneumonia in this patient. Authors reported that the percentages of pneumonia is the same (8/73 in the vaccinated group and 2/19 in the unvaccinated). However, these findings means that 73 vaccinated patients were admitted to the Hospitals against the 19 not vaccinated. Therefore, in the section Comparison of Unvaccinated and Fully Vaccinated BA.2-infected Patients who had COVID-19-typical Pneumonia authors should report and enhance that most patients with BA.2 and typical finding of COVID-19 pneumonia were vaccinated and only one vaccinated died. Therefore, this result should be enhance in this section because it represents a limits of actual m RNA

vaccines.

[Response]

As you pointed out, Case #8 showed atypical consolidation because the left lower lung was atelectatic. However, the right lung showed ground glass shadows, and based on the clinical course of the disease, we diagnosed the patient with COVID-19 typical pneumonia. This was noted in the Result section in page 18, line 281-284,

We also added about BA.2-infected patients with pneumoniae and limitation of mRNA vaccine the following text in this section

(page 20, line 318-322)

In this study, of the ten BA.2-infected patients with COVID-19-typical pneumonia, two were unvaccinated (11%, 2/19) and eight were fully vaccinated (11%, 8/73). Although the vaccination rate was 79% among those infected with BA.2 (Figure 2A), the number of patients with findings of COVID-19 typical pneumonia remained high, and one patient died (Figure 4).

(page 20, line 328-331)

These results suggest that while the vaccine may be effective against the spread of lung lesions, frail and elderly individuals with some comorbidities infected with BA.2 are at risk of developing pneumonia and mortality [30], suggesting limitations of vaccines designed based on the original strain.

[Comment]

Could these features be explained by the reduction of the immune response that was hypothesized by some authors especially with multiple doses of COVID-19 mRNA vaccines? Seneff, S., Nigh, G., Kyriakopoulos, A. M., & McCullough, P. A. (2022). Innate immune suppression by SARS-CoV-2 mRNA vaccinations: The role of G-quadruplexes, exosomes, and MicroRNAs. Food and Chemical Toxicology, 164, 113008.

I have already suggested this article but I don't find it in the discussion. Please discuss it

[Response]

In your suggested review by Seneff, S et al. (Food Chem Toxicol. 2022), the description of the immune response is explained by citing Mishra and Banerjea (Front Immunol. 2021). The content of the *in vitro* data was that ectopic overexpression of SARS-CoV-2 spike protein in the cell line releases large amounts of exosomes containing miR-148a and miR-590, which

suppress IRF9 and USP33. On this basis, they propose the idea that the anti-COVID-19 vaccines actively suppress type I IFN signaling because IRF9 is a key member of the IRF family of proteins involved in IFN- α activation. While such a plausible logical scenario may be possible, it is not yet well evidenced whether repeated vaccination actually suppresses the IFN signaling pathway. We consider that it is too early to discuss whether these events are physiologically occurring *in vivo* in people who have been repeatedly vaccinated.

【Comment】

I suggest to discuss also this article in the discussion: Tong X, Huang Z, Zhang X, et al. Old Age is an Independent Risk Factor for Pneumonia Development in Patients with SARS-CoV-2 Omicron Variant Infection and a History of Inactivated Vaccine Injection. Infect Drug Resist. 2022;15:5567-5573. doi:10.2147/IDR.S380005

【Response】

We cited the paper by Tong et al. in Discussion. We described in revised manuscript as follows.

(page 22, line 353-357)

Nevertheless, old age was identified as a risk factor for developing pneumonia in patients infected with the Omicron after receiving an inactivated vaccine injection [35]. This data is supported in the present study, which shows that older, high-risk patients with comorbidities who are infected with the Omicron BA.2 variants are still at risk of developing pneumonia and need early treatment to avoid severe disease.

【Comment】

Therefore, authors in the discussion should enhance also the limits of the actual COVID-19 vaccines that showed a high efficacy during the first phase of the pandemic. However, the short duration of the immune response after the vaccination and covid-19 pneumonia in the vaccinated patients were not reported in the clinical trials but we surprisingly found them in the clinical practice.

【Response】

As you commented, we discussed the following regarding the limitations of the COVID-19

vaccine.

(page 23, line 369-373)

The first developed vaccines were designed against the original virus and were highly effective in the early phase of pandemics. However, it should be noted that conventional vaccines may not adequately protect against infection or prevent severe disease because Omicron has numerous mutations in the S gene.

【Comment】

On the other hand, I think that with the omicron variant we can stop to describe the disease as COVID-19 because the disease is very different and less aggressive than the COVID-19 of the first phase of the pandemic. In fact, the unvaccinated group shows less frequent pneumonia than the vaccinated group and on imaging are more frequent atypical findings. Authors should describe better this finding in the discussion

These considerations are important because they would mean rethinking COVID-19 management policies.

I suggest also to report in the discussion the typical and atypical radiological findings of the pneumonia of COVID-19 in the first phase of the pandemic.

Therefore, I suggest a minor revision

It is a wonderful work because it is original, contains clinical and radiological findings and reflect the clinical practice.

Congratulations for this work

【Response】

We described in the paper the aggressiveness of the virus and the frequency of pneumonia cases during the first phase of the pandemic and now, as follows.

(page 21-22, line 348-351)

CT findings showed an organizing pneumonia pattern was typical in the early phase of the pandemic, but a bronchiolitis pattern was more common after the emergence of Omicron.

Both the incidence of pneumonia and CT score were lower in Omicron-infected patients, suggesting Omicron is less aggressiveness.

----- Responses to the comments of Reviewer #2 -----

【Comment】

The authors have satisfactorily responded to the reviewer comments, but still the manuscript does not have a novel finding and message for our readers and does not contribute to the literature significantly

[Response]

We thank you for reviewing our manuscript. Although this study may support previous data, we believe the novelty lies in the fact that we evaluated the pneumonia picture over time and the pneumonia picture of breakthrough infection.

----- Responses to the comments of Reviewer #3 -----

[Comment]

In this revision, the authors had added case numbers of BA.2 infection, and the data became much more persuasive. However, at line 406-413 in Conclusion, the opinions regarding booster vaccination were not properly addressed and should be remodified. The authors may discuss the concerns of repeated vaccination which could possibly cause “vaccine exhaustion” (according to the comments of reviewer#1), the potential benefits of next generation bivalent COVID-19 vaccines, and, the novel nasal spray COVID-19 vaccines, which have possible impacts on both reducing disease severity and viral transmission.

[Response]

We thank you for reviewing our manuscript. Regarding the booster vaccine, we consider that there is still a need for vaccination in high-risk patients, such as those with underlying diseases and the elderly. On the other hand, we clearly stated in the discussion that there are potential adverse effects of repeated vaccination. In the discussion, we also wrote about vaccine exhaustion, bivalent COVID-19 vaccines and the novel nasal spray as follow:

(page 25-26, line 415-426)

In the future, additional vaccination may be promoted preferentially in the older adult population and for those at high risk of disease progression owing to underlying diseases [46, 47]. However, repeated vaccinations have potential negative risks, such as vaccine exhaustion, adverse events, and vaccine resistance [48]. To be taken into consideration is the fact that the vaccines designed for original virus are already not compatible with the currently circulating Omicron variants because of a poor antigenic match. In that context, the Omicron BA.4/BA.5-adapted bivalent COVID-19 vaccines are expected to protect against Omicron subvariants and to have improved efficacy against severe disease [49]. In addition,

if intranasal vaccinations and nasal sprays become more developed, suppressing viral growth and clearing viruses in the nose and the upper respiratory system by triggering mucosal immune responses would deter viral spread [50-52].

【Comment】

In addition, the case numbers in figure legends of fig 4 (a) (b) and fig 5 (a) (b) were incorrect.

【Response】

We also thank you for pointing out the typo in Figure legends.

We corrected the case numbers in the Figure legends of fig 4 (a) (b) and fig 5.

----- Responses to the comments of Reviewer #4 -----

【Comment】

The authors have addressed all my comments and concerns sufficiently.

One minor comment:

The authors should use the unpaired Mann-Whitney test for their comparisons in Supplemental Figures 2C, 2D and 3 instead of the Wilcoxon test since the data are unpaired.

I don't think it'll change the results but the Wilcoxon test is inappropriate here.

【Response】

We thank you for reviewing our manuscript.

In the statistical analysis, we used unpaired and the Wilcoxon Rank Sum Test (also known as Mann-Whitney U test). Hence, there is no change in the statistical values. In the text, the term "Mann-Whitney U test" is used.

REVIEWERS' COMMENTS:

Reviewer #1 (Remarks to the Author):

The authors answered my questions and made an acceptable revision

Reviewer #3 (Remarks to the Author):

The authors have thoroughly addressed all my concerns and comments. Congratulations for this excellent work.

Reviewer #4 (Remarks to the Author):

In response to comments of reviewer #3:

I think it is inappropriate for the authors to use the term "vaccine exhaustion" or to cite reference 48, an opinion piece that does not actually provide evidence that repeated COVID vaccinations "exhausts" the immune system. That is a rather controversial term that is used by people with anti-vax sentiments. "Vaccine resistance" is also a bit of a misnomer, as vaccine recipients aren't becoming resistant to vaccines, it's just that the newer Omicron variants are becoming resistant to immune responses generated by the original vaccines (as the authors have discussed). I would suggest modifying that new sentence (Lines 417-419) and to avoid using the terms vaccine exhaustion or resistance. Otherwise, I am happy for this manuscript to be accepted.

----- Responses to the comments of Reviewer #1 -----

[Comment]

The authors answered my questions and made an acceptable revision.

[Response]

Thank you for your peer review and constructive comments.

----- Responses to the comments of Reviewer #3 -----

[Comment]

The authors have thoroughly addressed all my concerns and comments. Congratulations for this excellent work.

[Response]

Thank you for your peer review and constructive comments.

----- Responses to the comments of Reviewer #4 -----

[Comment]

In response to comments of reviewer #3:

I think it is inappropriate for the authors to use the term "vaccine exhaustion" or to cite reference 48, an opinion piece that does not actually provide evidence that repeated COVID vaccinations "exhausts" the immune system. That is a rather controversial term that is used by people with anti-vax sentiments. "Vaccine resistance" is also a bit of a misnomer, as vaccine recipients aren't becoming resistant to vaccines, it's just that the newer Omicron variants are becoming resistant to immune responses generated by the original vaccines (as the authors have discussed). I would suggest modifying that new sentence (Lines 417-419) and to avoid using the terms vaccine exhaustion or resistance. Otherwise, I am happy for this manuscript to be accepted.

[Response]

Thank you for your comments. I agree with your suggestions and would like to delete the

sentence citing reference #48. The deleted sentence is as follows, “However, repeated vaccinations have potential negative risks, such as vaccine exhaustion, adverse events, and vaccine resistance [48].”.